# Analysis and control of untemplated DNA polymerase activity for guided synthesis of kilobase-scale DNA sequences

Simeon. D. Castle[1,7], Thea C. T. Irvine[1,7], Adrian Woolfson[2], Gregory Linshiz[2], Blake T. Riley [2], Ifor D. W. Samuel [3], Loren Picco[4], Philipp Holliger [5], Lauren M. Oldfield[2], Andrew Hessel[6] & Thomas E. Gorochowski [1] ✉

DNA polymerases are complex molecular machines capable of replicating genetic material using a template-driven process. While the copying function of these enzymes is well established, their ability to perform untemplated DNA synthesis is less well characterized. Here, we explore the ability of DNA polymerases to synthesize DNA fragments in the absence of a template. We use long-read nanopore sequencing, real-time fluorescence assays, and atomic force microscopy to observe the synthesis and physical structure of pools of DNA products derived from a diverse set of natural and engineered DNA polymerases across varying temperatures and buffer compositions. We detail the features of the DNA fragments generated, enrichment of select sequence motifs, and demonstrate that the sequence composition of the synthesized DNA can be altered by modifying environmental conditions. This work provides extensive data to better discern the process of untemplated DNA polymerase activity and may support its potential repurposing as a technology for the guided synthesis of DNA sequences on the kilobase-scale and beyond.

Genetic information encoded in DNA sequences is a defining feature of all life on Earth. It provides the instructions that cells require to synthesize and assemble their molecular machinery, encodes signals to coordinate cellular processes, and is the substrate for the process of evolution by natural selection. How biological information emerges de novo, however, remains unclear.

DNA polymerases (DNAPs) are typically viewed as molecular machines involved in the high fidelity copying of genetic information through the replication of DNA using a template-mediated process[1]. However, it was shown in the 1960s that some DNAPs have the ability to synthesize single-stranded DNA fragments in the absence of a pre-existing template[2–4]. This ab initio or "doodling" activity is widespread among polymerases[5] and has been observed in RNA polymerases,

notably T7 RNA polymerase[6], Qbeta RNA-dependent RNA polymerase[7] and 5TU RNA polymerase ribozyme[8]. Doodling suggests a mechanism for how de novo untemplated genetic information may be generated.

Investigations of this process have demonstrated that DNAPs have an intrinsic capacity to doodle[9–12], that this process is sensitive to environmental factors such as temperature and variations in buffer conditions[2,9,10,13], and that pools of synthesized DNA molecules produced in the absence of template likely go through a multi-stage process of semi-random synthesis and enrichment via the selection of sequences able to more efficiently self-replicate[5,14–16]. However, all such studies have been hampered by the inability to adequately characterize the pools of DNA molecules generated. Recent advances in the ability to read full-length DNA molecules using long-read sequencing

[1]School of Biological Sciences, University of Bristol, 24 Tyndall Avenue, Bristol, UK. [2]Replay Holdings Inc, 5555 Oberlin Drive, Suite 120, San Diego, CA, USA. [3]Organic Semiconductor Centre, School of Physics and Astronomy, University of St Andrews, North Haugh, St Andrews, UK. [4]Interface Analysis Centre, HH Wills Physics Laboratory, Tyndall Avenue, Bristol, UK. [5]MRC Laboratory of Molecular Biology, Francis Crick Avenue, Cambridge Biomedical Campus, Cambridge, UK. [6]The Center of Excellence for Engineering Biology, 33 Park Place, Suite 191, New York, NY, USA. [7]These authors contributed equally: Simeon. D. Castle, Thea C. T. Irvine. ✉e-mail: thomas.gorochowski@bristol.ac.uk

have made it feasible to generate near-complete sequence information from such vast pools, offering us insights into the generative process.

In this work, we employ nanopore sequencing, real-time fluorescence assays, and atomic force microscopy to provide the most comprehensive characterization to date of doodling kinetics, physical structure and sequence features of the full-length untemplated DNA molecules synthesized (Fig. 1a). We document the doodling function of a diverse array of natural and engineered DNAPs, assess how temperature and buffer composition impacts the nature of the sequences generated, and explore how this process may potentially be manipulated through limiting deoxynucleotide triphosphate (dNTP) availability. We identify considerably greater diversity in the features of the sequences generated than previously reported, document the complexity of the DNA molecules synthesized, and demonstrate that in select cases, modification of environmental conditions enables robust and directed changes to the sequences produced. In addition to providing a rich data set for further characterizing the nature of untemplated DNA synthesis, our findings also suggest a need to reconsider the role of DNAP function in cellular biochemistry. Overall, the work offers a perspective on how DNAPs may contribute to the genesis and evolution of new genetic information and may provide the basis of future methods that repurpose DNAPs as programmable machines for the synthesis of long DNA sequences.

## Results

### Single molecule analysis of doodled DNA

We began by focusing on Taq and Vent polymerases that have been shown to be capable of untemplated DNA synthesis (Fig. 1b)[9,12]. Isothermal reactions were established using standard buffers for each polymerase at 65 °C and 74 °C and run for 16 h (**Methods**). We then performed nanopore sequencing of each reaction to characterize the sequences present in the pools of newly synthesized DNA fragments. To ensure that the reads containing "doodled" products were accurate, we trimmed sequencing adapters from the ends of each read, filtered out concatemer reads (namely those containing sequencing adapters within the read itself that likely occur due to ligation of separate fragments during the preparation of the sequencing libraries), and performed read filtering based on a median Q score ≥13 and length >65 nt to remove sequences that might correspond to DNA fragments

introduced when preparing the sequencing libraries (i.e., adapter and barcode fragments) (Methods).

Analysis of the read length distributions showed substantive differences for each of the polymerases (Fig. 2, top sections of each panel). Taq displayed a bimodal distribution with peaks at ~80 nt and 400 nt for reactions at 65 °C, with a shift in the peak at 400 nt to 500 nt when the reaction temperature was increased to 74 °C. This distribution also displayed a much longer tail (Fig. 2a, b), with over 20% of reads >1000 nt compared to only 4–7% at 65 °C (Table 1). For both temperatures, similar concentrations of synthesized DNA were measured (~1000 ng/μL). In contrast, Vent had only a single peak in the read length distributions at ~80 nt, with a rapid exponential-type decay as read length increased that was similar across temperatures and which resulted in only a very small fraction of reads being >250 nt long (Fig. 2c,d). The differences in the shape of the read length distributions suggest that there may be factors (e.g., the formation of DNA secondary structures) that hamper Vent's ability to synthesize long DNA fragments. Additionally, ~10-fold less DNA was produced at the lower temperature and only small fractions of reads were >1000 nt long (≤ 6%) across both temperatures. This suggests Taq is more efficient at untemplated DNA synthesis and able to produce longer DNA fragments than Vent, and that higher temperatures are more conducive to the synthesis of longer untemplated DNA synthesis.

The nanopore sequencing additionally captured information regarding sequence composition and characteristic repetitive patterns (i.e., sequence motifs) within each DNA fragment. To characterize these features, we performed a number of linked analyses to generate visualizations of the DNA pools where each analysed read is denoted by a vertical column across all the plots (Fig. 2, lower heatmaps). Analyses included assessing the base composition (red heatmaps) for each read, the transition probabilities from one base to another (blue heatmaps), and the sequence autocorrection to extract the scale (i.e., length/lag) of any repetitive sequences present (blue to yellow heatmaps). Each of these analyses was performed for a subset of reads shorter and longer than 250 nt and the results hierarchically clustered to ensure similar reads were grouped in the plots as vertical columns. The top seven clusters were highlighted to simplify their classification (alternating dark and light gray sections at the bottom of the panels). From these visualizations, it was possible to assess distinctive features

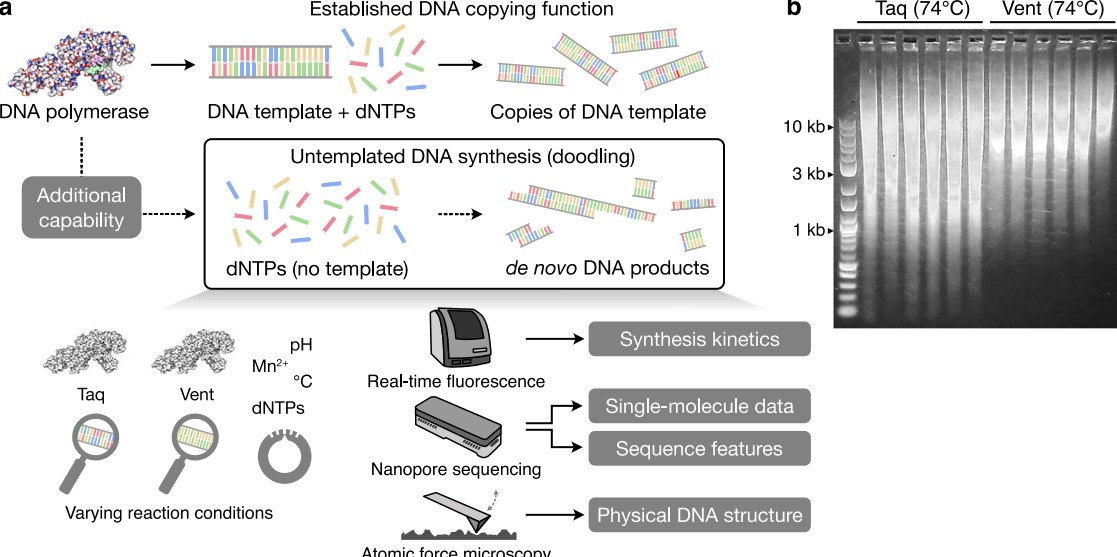

**Fig. 1 | Methodology to study doodling by DNA polymerases. a** Overview of the established copying function of DNA polymerases and the less well studies capability for them to perform untemplated DNA synthesis. In addition, methods used to study the process and the key data generated in this work are shown. **b** Varying

length distribution of DNA doodled by Taq and Vent polymerases after 16 h at 74 °C with no DNA template present. For each polymerase, 6 experimental replicates are shown.

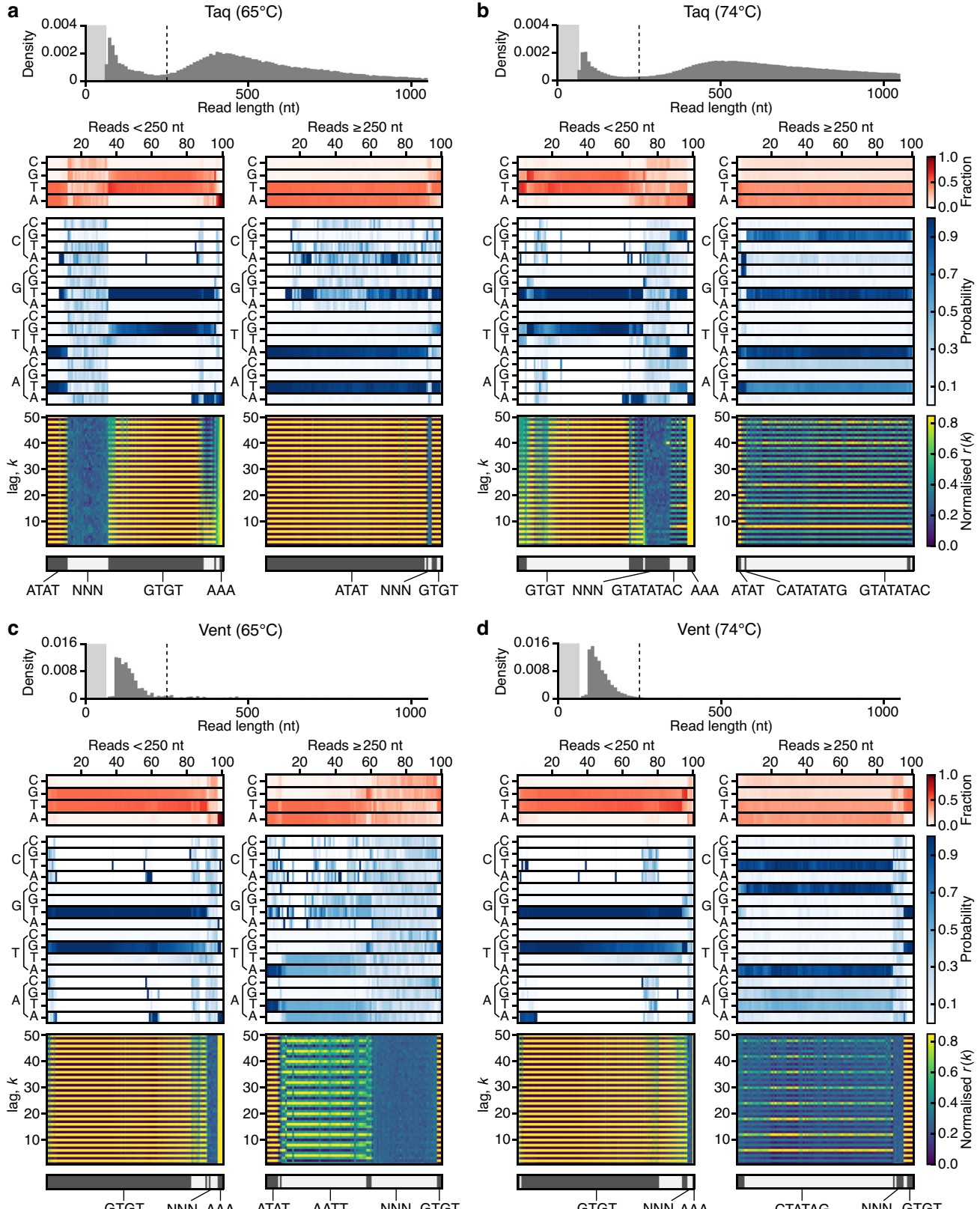

**Fig. 2 | Single molecule analysis of doodling activity for Taq and Vent polymerases at different temperatures. a** Taq at 65 °C. **b** Taq at 74 °C. **c** Vent at 65 °C. **d** Vent at 74 °C. In all panels, the top histogram shows the sequence length distribution with the lightly shaded region denoting the 0–65 nt range and dashed line denoting the 250 nt read length. Below this, heatmaps show for a random subset of reads smaller and larger than 250 nt (left and right plots, respectively) the following information (top–bottom): 1. Sequence composition (red heatmap), 2. Probability of transitioning from one base to another (blue heatmap), 3. Autocorrelation analysis capturing the similarity of the sequence compared to itself after varying nucleotide shifts/lag $k$ (blue to yellow heatmap), and 4. the seven top clusters of reads (alternating light and dark gray) with key clusters having the sequence repeat they include below. Reads are displayed vertically and hierarchically clustered such that similar sequences are grouped.

**Table 1 | Doodling activities of DNA polymerases [a]**

| Polymerase | Temp. (°C) | Buffer | Additives | DNA mass [b] (ng/µL) | % Reads > 250 nt | % Reads > 1000 nt |
|---|---|---|---|---|---|---|
| Taq | 74 | Taq Standard | – | 856 | 88.0 | 22.9 |
|  |  |  |  | 1340 | 88.1 | 25.2 |
|  |  | Minimal pH 8.2 | – | 195 | 87.4 | 4.9 |
|  |  |  |  | 136 | 68.5 | 5.2 |
|  | 65 | Taq Standard | – | 1140 | 81.5 | 7.4 |
|  |  |  |  | 1000 | 53.4 | 4.3 |
|  |  | Minimal pH 8.2 | – | 31 | 93.5 | 0.4 |
|  |  |  |  | 18 | 93.6 | 0.5 |
|  |  | Minimal pH 9.5 | – | 24 | 88.0 | 0.5 |
|  |  |  |  | 32 | 95.5 | 0.8 |
|  |  | Taq Standard | MgCl$_2$ (25 mM) | 912 | 74.8 | 5.7 |
|  |  |  |  | 972 | 35.2 | 2.6 |
| Vent | 74 | Thermopol | – | 1220 | 9.8 | 3.3 |
|  |  |  |  | 1380 | 13.9 | 6.0 |
|  | 65 | Thermopol | – | 120 | 16.4 | 2.0 |
|  |  |  |  | 118 | 5.5 | 0.7 |
| Vent exo– | 65 | Thermopol | – | <4 | 2.1 | 0.0 |
|  |  |  |  | <4 | 1.8 | 0.1 |
| Therminator | 65 | Thermopol | – | <4 | 2.2 | 0.1 |
|  |  |  |  | <4 | 1.1 | 0.0 |
| RT521K | 65 | Thermopol | MnCl$_2$ (25 mM) | <4 | 28.5 | 1.2 |
|  |  |  |  | <4 | 26.1 | 0.7 |
| 3A10 | 65 | Taq Standard | MgCl$_2$ (25 mM) | <4 | 2.2 | 0.1 |
|  |  |  |  | <4 | 4.9 | 1.1 |

a.All reactions run for 16 h with DNA mass and read lengths produced given separately for both experimental replicates.
b.Values of <4 ng/µL were below the detection limit of the fluorometer used.

of each doodled DNA sequence and to make comparisons between samples.

Using this approach, the Taq polymerase reactions at 65 °C showed a majority of GT repeats (61%) for reads <250 nt, in addition to less frequent AT repeats (13%), poly-A sequences (4%), and random sequences (23%) (Fig. 2a). In contrast, longer reads at this temperature were nearly exclusively AT repeats (91%), with only a small fraction of GT repeats and random sequences (9% in total). At 74 °C, there was a distinctive shift in the sequence composition, with smaller reads retaining the dominance of GT repeats (71%), and longer reads demonstrating the emergence of GTATATAC repeats (93%) and a small fraction of CATATATG (3%) repeats. The increased fraction of reads >1000 nt at 74 °C, may relate to the fact that these longer repeats (when found in multiples of 3 or more) can efficiently self-extend by looping and base pairing of the complementary repeats and extension by Taq, using one or more of the single stranded repeats as a template[5,14,15].

For the Vent polymerase (Fig. 2c, d), there was again a dominance of repetitive GT sequences (91–96%) and a small fraction of poly-A sequences (1–3%) for reads <250 nt across both temperatures. However, for longer reads >250 nt, there were large differences across temperatures as compared with Taq, with Vent reactions at 65 °C showing only a small number of repetitive GT and AT sequences, Instead, reads were dominated by AATT repeats (48%) and random sequences (37%) with a near even coverage of bases (Fig. 2c). This was substantially different to the AT repeat dominated Taq results (Fig. 2a). Interestingly, the Vent reactions at 74 °C produced a small fraction of longer reads containing GT repeats and random sequences (12% in total). However, the vast majority (88%) contained CTATAG repeats (Fig. 2d). This again, differed from the longer GTATATAC repeat observed for the Taq reactions under similar conditions, and highlights potential biases in the affinities these polymerases have for the addition of specific nucleotides. At 74 °C, the longer reads >250 nt demonstrated that Vent favors C → T, G → C, T → A and A → G or T, while Taq has a strong preference for C → G, G → T, T → A and A → T transitions. Furthermore, these

preferences were influenced by temperature and appear to impact the efficiency with which DNA synthesis occurs (different transitions dominate the analysis of the short and long fragments).

Overall, these results were found to be robust for both polymerases tested and across different temperatures, with similar patterns observed in biological replicates (Supplementary Figs. 1–4).

### Assessing the capture of high molecular weight DNA

One of the major challenges when working with high molecular weight (HMW) DNA for nanopore sequencing is the chance of shearing during the preparation of sequencing libraries. This can lead to smaller than expected products. To assess whether this might be the case for our results, which are generated from barcoded sequencing libraries to allow for the assessment of many different polymerases and conditions within a single sequencing run, we carried out an adapted sequencing protocol designed to help capture full length HMW DNA molecules. This was performed for doodling reactions containing the Taq polymerase with the Taq Standard buffer at 74 °C (Methods).

Comparing the read length distributions of the barcoding and HMW protocols, we observed a good reproducibility in the shape of the distributions across replicates (Supplementary Fig. 5). The most prominent difference was that there were virtually no reads 65–250 nt long for the HMW protocol (<1%, Supplementary Fig. 5b), whereas -12% for the barcoding protocol (Supplementary Fig. 5a). This difference likely stems from the barcoding requiring additional bead-based clean-up steps that could lead to minor shearing of the DNA and the generation of smaller length fragments. In terms of longer reads, >99% were longer than 250 nt and >35% longer than 1000 nt for the HMW protocol, demonstrating its ability to capture long reads. In contrast, the barcoding protocol saw an -11% fewer reads both >250 nt and >1000 nt long compared to the HMW protocol. Importantly, most of the very large fragments observed for the HMW protocol were between 250 nt and 2000 nt, which were well captured by the barcoding protocol. Therefore, while some very long sequences may not be captured by the barcoding approach, it is still able to provide

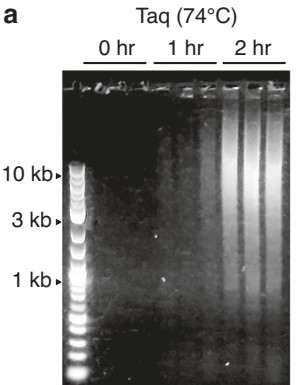

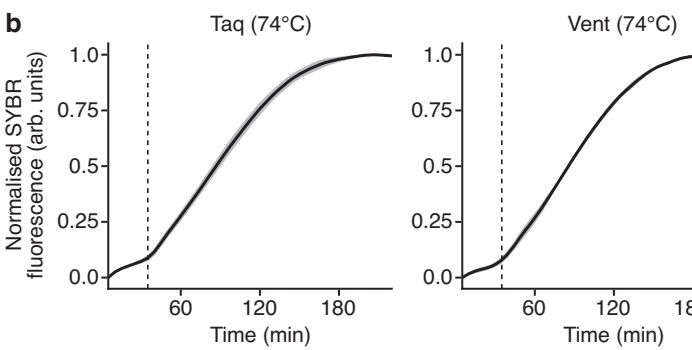

**Fig. 3 | Kinetics of doodling activity for the Taq and Vent polymerases.**
**a** Emergence of doodled DNA products during an isothermal reaction at 74 °C with Taq polymerase over 2 h. For each timepoint, 3 experimental replicates are shown (grouped lanes). **b** Normalized SYBR fluorescence for the Taq (left) and Vent (right) polymerases during the isothermal reactions in arbitrary units (arb. units). Data normalized by the maximum SYBR fluorescence of each sample. Solid black line shows the average of the 4 experimental replicates and gray shaded region denotes ±1 standard deviation. Thin dashed line denotes the 30 min point in the reaction.

insights into the majority of types of molecule produced from a doodling reaction, and ensures representative read length distributions.

### Kinetics of doodling activity

Historically, the dynamics of doodling has principally been studied by time-sampling reactions and visualizing DNA products by gel electrophoresis (Fig. 3a)[17]. To more precisely quantify the doodling kinetics, we used a real-time fluorescence assay to track the formation of DNA molecules synthesized under isothermal conditions. The experiments were run for 16 hours at 74 °C for both the Taq and Vent polymerases. In both cases, we found that doodling occurred in two distinct stages (Fig. 3b). Initially, a slow increase in synthesized DNA was observed. Then, at around 30 min, a transition to a second phase occurred with a 3–3.5-fold increase in DNA synthesis rate. This rate was fixed (there was a linear increase in DNA mass) for a further 1.5 h, after which the rate steadily decreased until DNA synthesis stopped 3 h following the initiation of the reaction. This second stage appeared to capture a saturation in the ability for DNA synthesis, likely due to the depletion of free dNTPs or production of DNA products that cannot be further extended. This behavior was consistent across replicates, with similar transition points and DNA synthesis rates (Fig. 3b).

This two-stage process is consistent with a previously suggested mechanism of untemplated DNA synthesis, whereby an initial stage allows for the generation of short random DNA sequences that may subsequently by chance discover repetitive sequences able to more efficiently self-replicate in a second stage[5,14,15] (mentioned previously). These results therefore further support the role of self-replicating sequences in the generation of new genetic material during the doodling process.

### Influence of environmental factors

It has been demonstrated that environmental factors such as temperature and buffer composition can influence the length and sequence composition of doodled DNA[2,9,10,13]. While we have shown the important role that temperature plays in determining sequence composition (Fig. 4), to further characterize this effect, we studied the doodling activity of Taq polymerase across a range of buffers. These included a minimal buffer with varying pH (8.2 and 9.5) and a standard Taq buffer including excessive amounts of $MgCl_2$, which is known to stabilize the annealing of primers to incorrect template sites when used in PCR reactions and thereby decreasing the specificity of the amplification process.

Similar to previous experiments, the results were shown to be highly reproducible for many of the conditions, with similar patterns observed across experimental replicates (Supplementary Figs. 6–9). The main exceptions were Taq at 65 °C in Taq standard buffer supplemented with 25 mM $MgCl_2$ and Taq at 74 °C in minimal buffer at pH 8.2. Replicates under these conditions resulted in similar qualitative features for the read length distributions, but quantitative differences in the ratio of short and long fragments. A comparison of the sequence composition and repetitive motifs across different conditions showed that in most cases the same patterns were maintained. The only notable differences were for Taq at 65 °C when using a minimal buffer at both pH 8.2 and 9.5, which displayed a slight decrease (>2-fold) in the fraction of GC repeats for reads <250 nt (Fig. 4b, c), and the more complex sequence repeats produced by Taq at 74 °C in the minimal buffer at pH 8.2 (Fig. 4a), which for one of the replicates (replicate 1) also contained longer repetitive sequences at smaller (<250 nt) read lengths (Supplementary Fig. 9).

The most notable differences observed were in the amount of DNA synthesized and the fraction of long (>1000 nt) DNA sequences produced (Table 1). Most of the reactions saw a decrease in the amount of DNA synthesized (apart from those at 65 °C in the Taq Standard buffer with $MgCl_2$), with the minimal buffer leading to an ~10-fold decrease at 74 °C and ~100-fold decrease at 65 °C. Smaller drops in the percentage of long reads >1000 nt were also seen, as well as differences in the shape of the read length distributions. This suggests underlying changes in the DNA synthesis process. Specifically, reactions that used the minimal buffer at 65 °C with a pH of 8.2 or 9.5 resulted in far fewer small DNA fragments <250 nt, and read length distributions with a symmetric Gaussian-like shape (Fig. 4b, c). This would be indicative of a generative process whereby all fragments in the pool have similar rates of synthesis, but where there is some stochasticity in the process to cause variation at the pool level. Interestingly, these reactions were dominated by reads containing AT repeats, with little diversity in the repetitive motifs present (especially for longer reads >250 nt). Given the amplification of different motifs is likely to occur at different rates, having only a single dominant motif may account for the simpler read length distribution in these specific conditions (i.e., it imposes specific constraints on the typical maximum length that can be achieved).

Together, these results highlight the role that environmental factors play in biasing the sequence composition and affecting the synthesis efficiency of the doodling process.

### Impact of cycling thermal conditions

All experiments so far were carried out under isothermal conditions. As the amplification of the repetitive sequences would require binding and unbinding of the ssDNA molecules to proceed efficiently, we

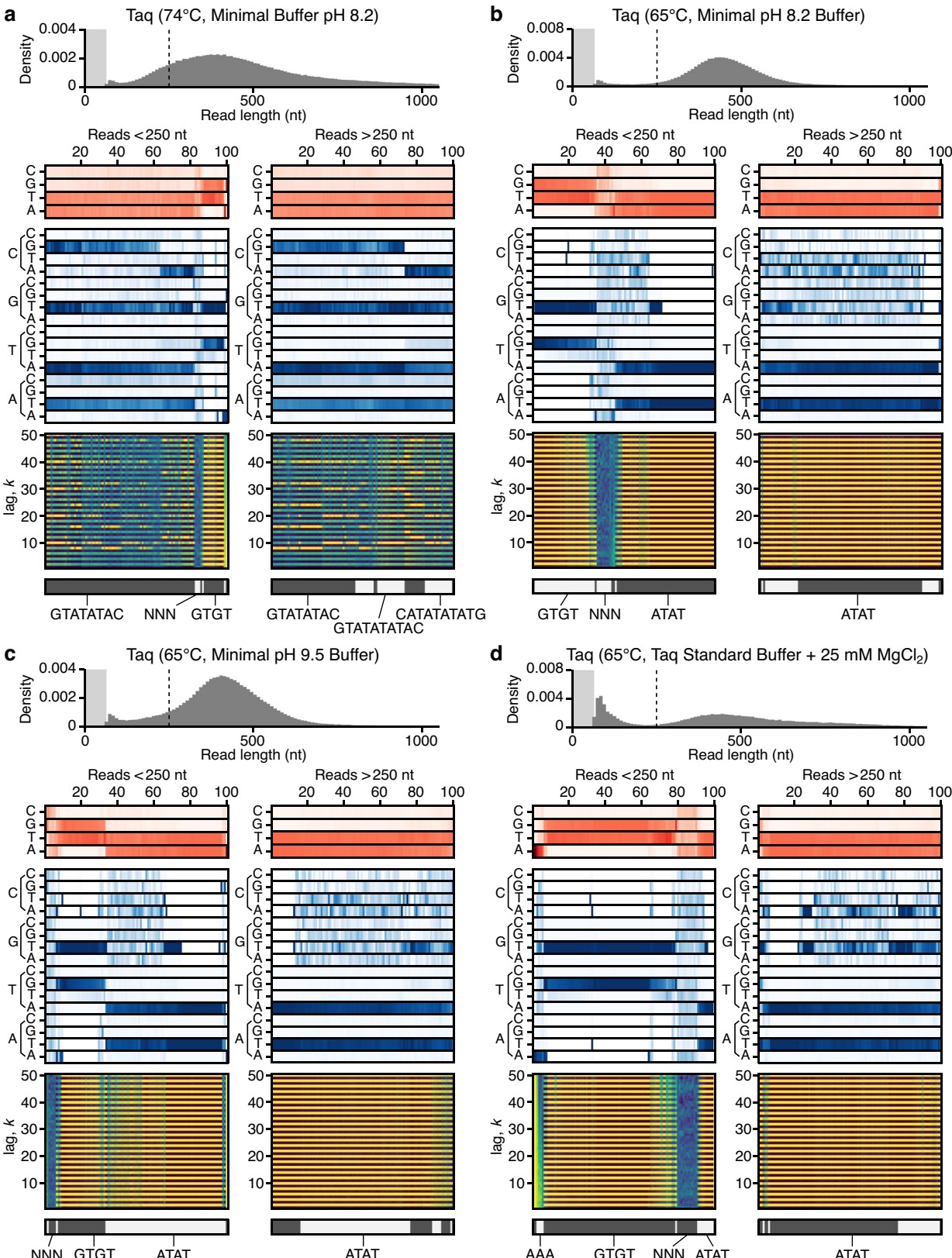

wondered whether thermal cycling might enhance this process. To test this hypothesis, doodling reactions for both Taq and Vent were established where the temperature was cycled 9 times between 65 °C for 20 mins and 74 °C for 20 mins, for a total reaction time of 6 h. DNA from these reactions was then nanopore sequenced and analyzed in a similar way as for the isothermal reactions.

Counterintuitively, we found that for both Taq and Vent, temperature cycling virtually eradicated the occurrence of repetitive sequences, with only a small fraction (<2.5%) showing a repetitive sequence motif (Supplementary Fig. 10a). Interestingly, at least an order of magnitude less DNA was produced in these reactions compared to the normal isothermal doodling reactions (<12% compared

**Fig. 4 | Single molecule analysis of doodling activity for Taq under different environmental conditions. a** Taq at 74 °C in minimal buffer at pH 8.2. **b** Taq at 65 °C in minimal buffer at pH 8.2. **c** Taq at 65 °C in minimal buffer at pH 9.5. **d** Taq at 65 °C in Taq standard buffer with 25 mM MgCl₂. In all panels, the top histogram shows the sequence length distribution with the lightly shaded region denoting the 0–65 nt range and dashed line denoting the 250 nt read length. Below this, heatmaps show for a random subset of reads smaller and larger than 250 nt (left and right plots, respectively) the following information (top–bottom): 1. Sequence composition (red heatmap), 2. Probability of transitioning from one base to another (blue heatmap), 3. Autocorrelation analysis capturing the similarity of the sequence compared to itself after varying nucleotide shifts/lag $k$ (blue to yellow heatmap), and 4. the seven top clusters of reads (alternating light and dark gray) with key clusters having the sequence repeat they include below. Reads are displayed vertically and hierarchically clustered such that similar sequences are grouped.

to Taq and Vent at 74 °C, and 1% compared to Vent at 65 °C), and there was a near uniform transition from one base to another in the sequences produced (Supplementary Fig. 10b). More detailed analysis showed that all samples were dominated by sequences that were a mixture of contaminants, mostly from the *Escherichia coli* genome (Supplementary Data 1). Partial sequences were commonly observed for the uncharacterized *ydfO*, *ybcW* and *ylcI* genes related to the DLP12 and Qin prophage. DNA fragments containing these sequences were likely carried through from the purification of the enzymes by the manufacturers and the cycling of temperature during the reactions may have allowed for non-specific binding and extension to create the small number of chimeric contaminant sequences seen. While commercial providers do guarantee only trace amounts of such contaminants in their products (≤1 genome per μL from quality control assessments using qPCR by the commercial providers of the reagents used in this study), the sensitivity of the nanopore sequencing enables them to be picked up when virtually no doodling occurs.

To assess whether these contaminants might confound the results of other doodling experiments, we analyzed the *E. coli* BL21(DE3) genome to see whether the repeats we observed in our doodling reactions that are able to self-amplify could have been generated by template-mediated synthesis from genomic contaminants (Methods). Only a single occurrence of a repeated GATATC motif was found. This makes it highly unlikely that genomic DNA contaminants from the enzyme purifications are the cause of the diverse and highly repetitive sequences that dominate the doodling reactions. Furthermore, the very small amounts of contaminants in the isothermal doodling reactions, suggests that amplification of these products is unable to proceed efficiently, unlike the doodling process.

## Physical structure of doodled DNA pools

Sequencing of doodled DNA products provides some insight into the composition of the DNA produced, but is unable to capture the physical structures these molecules produce individually and through interactions with other molecules within the pool. To assess these features, we performed atomic force microscopy (AFM) for the Taq polymerase reactions. We used a high-speed contact mode AFM approach, which allowed us to image large-scale structures (areas of 1.3 μm² per image) at a sub-nanometer resolution (Fig. 5a; Methods).

From the AFM images we were able to extract 788 molecules at sufficient contrast to accurately evaluate their structure. It was immediately evident that a large proportion of the molecules displayed a clear branching structure (Fig. 5b) with 23% containing one or more branches. In addition, there was a steep drop off in the number of branches per molecule and a maximum of 8 branches seen in a few cases (Fig. 5c). Given that ssDNA molecules are synthesized, these branches likely occur either at common repeat sequences found across different molecules or by a single molecule folding back on itself at repetitive, palindromic sequences to create a tight hairpin with a non-binding end that is observed as a separated branch. Closer inspection of some of these imaged molecules showed increased peaks in height at points where branches occurred (Fig. 5b, white arrows). This suggests that these regions may be double stranded, supporting the idea of base pairing between separate ssDNA molecules, or through the formation of hairpins within a single ssDNA molecule.

The ability to view the DNA molecules allowed us to calculate physical lengths that could then be converted into numbers of nucleotides using the fact that a single nucleotide is 0.34 nm long[18] (Fig. 5d). We calculated median DNA molecule lengths of 518 nt and 552 nt from the AFM and nanopore sequencing data, respectively, for the same Taq doodling reactions at 74 °C with the Taq Standard buffer. The longest molecule we imaged was estimated to be 2769 nt long. The small difference of ~6% in the median molecule lengths lends further support to the nanopore sequencing recovering accurate lengths of the molecules present in these reactions.

As well as providing support for the accuracy of the sequencing data, the branched structure of many of the doodled molecules could also provide an explanation for the very long DNA molecules (>10,000 nt) inferred from the electrophoresis gels (Fig. 1b and Fig. 3a), but not observed at such high levels in the sequencing data or via AFM. The reason could be that rather than there being very long ssDNA molecules present, instead the ability for the ssDNA sequences to base pair with each other through the common repetitive motifs would enable the creation of very large multi-molecule structures that would run much slower through a gel than a single-molecule counterpart. This would confound the typical interpretation of these gels and further highlights the importance of performing complementary measurements like sequencing and AFM to better understand poorly understood biological processes.

## Doodling activity of other DNA polymerases

Given the distinct sequence compositions of the doodled products derived using Taq and Vent polymerases, it appears likely that other DNAPs may have unique biases in the untemplated sequences they generate. To further examine this, we considered two other DNAPs: Vent exo− to determine whether mutations in Vent that remove its proofreading exonuclease activity impact doodling, and Therminator as it was originally sourced from a distantly related species (*Thermococcus* species 9°N-7) and is known to incorporate modified substrates. In addition, we also studied two engineered DNAPs, RT521K and 3A10[19,20]. RT521K is an extensively engineered variant of Tgo polymerase[19] and 3A10 is a chimera of Tth and Taq polymerase generated by molecular breeding and compartmentalized self-replication (CSR) selection[20]. We carried out 16 h isothermal reactions and performed nanopore sequencing to characterize the sequences of the doodled DNA products (Methods).

Highly reproducible results were seen for all four polymerases, with read length distributions displaying an exponential decay for longer read lengths, similar to previous experiments with Vent (Supplementary Figs. 11–14). For Vent exo−, we found large fractions of reads containing GT repeats (93% of reads <250 nt), a lack of the more complex AATT repeat sequences seen with standard Vent polymerase, and an increase in the fraction of reads with near-random sequences (65% of reads >250 nt) (Fig. 6a). Therminator displayed similar features, but contained a small number of longer reads (>250 nt) with more complex sequence repeats, including CTATAG, GTATATAC and AATT (3–8% total across replicates) (Fig. 6b). For the engineered polymerases, we found that the short reads of RT521K were once again dominated by GT repeats, but virtually all longer reads >250 nt had random-like sequences (97%) (Fig. 6c). This was evident from the near-equal fraction of bases and transition probabilities between bases. The

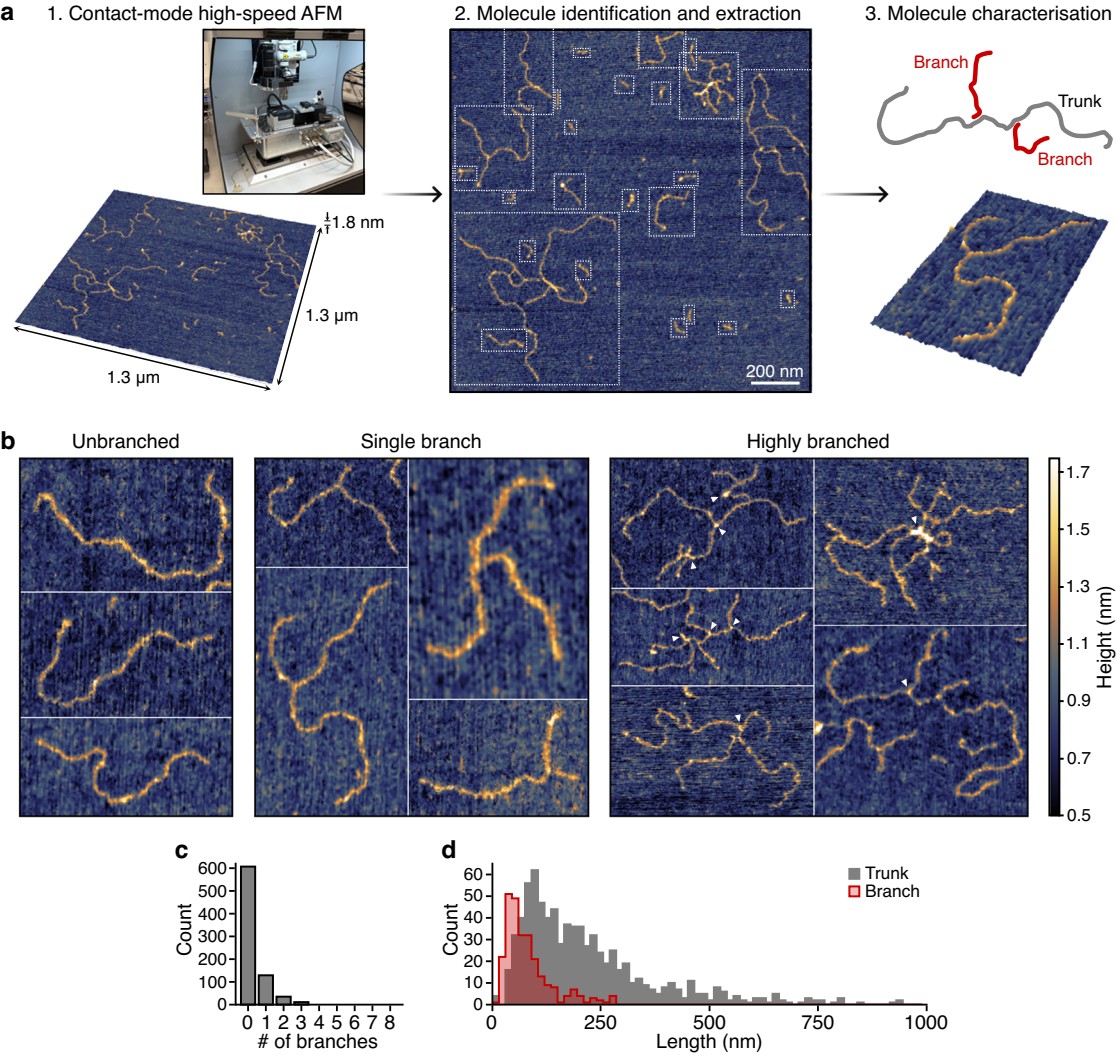

**Fig. 5 | Imaging of doodled DNA using atomic force microscopy. a** Images of doodled DNA products were collected over large areas (1.3 μm²) using contact-mode high-speed atomic force microscopy (AFM). From these full frame images, individual molecules were identified and extracted using a molecule identification algorithm (white dashed boxes) and each characterized in terms of the estimated longest continuous strand, which we call the 'trunk', as well as the length of separate 'branches' that stem from this. Length of each trunk and its associated branches were calculated and stored as a group for further analysis and considered a single element. A schematic of the molecular structure seen is shown in the top right of the panel with the trunk in gray and branches in red. **b** Representative extracts from AFM images showing unbranched, single branch and highly branched molecules. White arrows in highly branched images highlight raised areas near branch points and at the ends of molecules (e.g., points of expected hairpin formation). For every image, each pixel is ·0.48 × 0.48 nm in size. **c** Molecule counts of trunks containing different numbers of branches. **d** Distribution of molecule lengths extracted from AFM images.

3A10 polymerase showed a similar dominance of random-like sequences, although to a lesser extent than RT521K (Fig. 6d). When MnCl₂ and MgCl₂ were added to the buffers for the RT521K and 3A10 reactions, respectively, the probability of a shift to more random doodling became greater. For MnCl₂, this is likely due to the additive promoting the misincorporation of nucleotides and destabilizing existing biases in base transitions[21]. The most distinctive difference of these studied polymerases compared to Taq and Vent was the heavily reduced overall quantity of DNA synthesized per reaction ( < 4 ng/μL in all cases) and a lower fraction of reads >250 nt long (Table 1).

## Analysis of DNA polymerase transition frequencies

The polymerases considered so far span both Family A and B, which are known to have structural and functional differences that affect their mutational biases[22]. To assess whether these biases might underpin some of the favored transitions seen in the doodled DNA products, we generated averaged base transition probabilities for each DNAP and condition, grouping the polymerases by Family or classifying them as an engineered variant (Fig. 7a). From these transition matrices, it was clear to see common Family related biases. Specifically, in the doodled DNA products, all Family A DNAPs had a strong preference for A → T, T → A and G → C transitions, and a weak preference for C → A transitions. Furthermore, a change in temperature to 74 °C leads to the emergence of an additional strong preference for C → G transitions. Family B DNAPs show a different pattern, with a strong preference for G → T, T → G transitions and more even transitions from A and C nucleotides to all others at 65 °C, and the emergence of a strong C → T transition at 74 °C for Vent. These trends are mirrored for the 3A10 engineered polymerase, which is a chimeric variant made from two Family A DNAPs (Tth and Taq). Interestingly, the RT521K polymerase, which is from Family B and is the most heavily engineered of those studied, showed fewer overall biases. This leads to near even transition probabilities between all bases in the doodled DNA products and is also evident in the fewer repetitive motifs seen in the sequence pools.

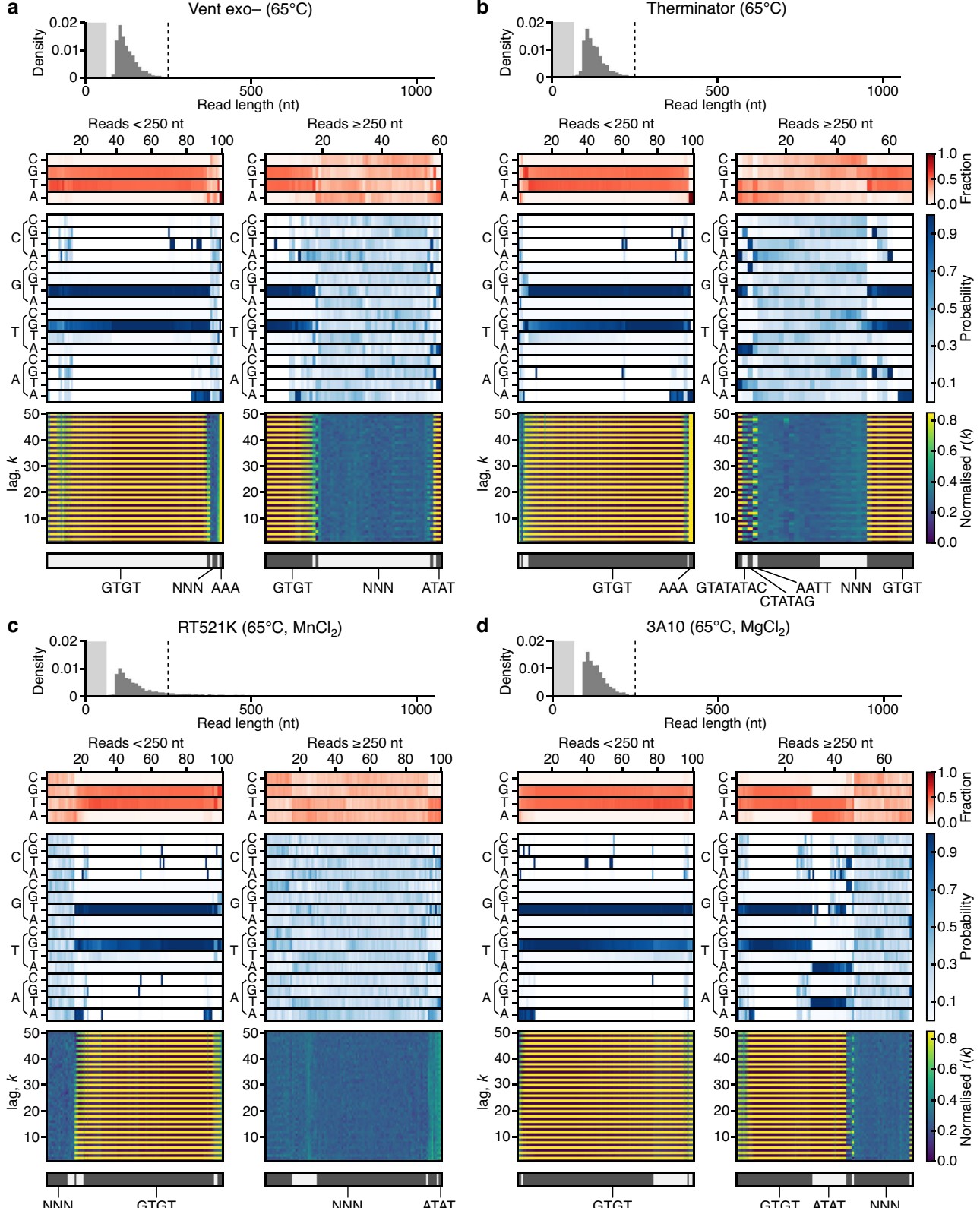

**Fig. 6 | Single molecule analysis of doodling activity for diverse DNA polymerases. a** Vent exo– at 65 °C. **b** Therminator at 65 °C. **c** RT521K at 65 °C with 25 mM MnCl₂. **d** 3A10 at 65 °C with 25 mM MgCl₂. In all panels, the top histogram shows the sequence length distribution with the lightly shaded region denoting the 0–65 nt range and dashed line denoting the 250 nt read length. Below this, heatmaps show for a random subset of reads smaller and larger than 250 nt (left and right plots, respectively) the following information (top–bottom): 1. Sequence composition (red heatmap), 2. Probability of transitioning from one base to another (blue heatmap), 3. Autocorrelation analysis capturing the similarity of the sequence compared to itself after varying nucleotide shifts/lag $k$ (blue to yellow heatmap), and 4. the seven top clusters of reads (alternating light and dark gray) with key clusters having the sequence repeat they include below. Reads are displayed vertically and hierarchically clustered such that similar sequences are grouped.

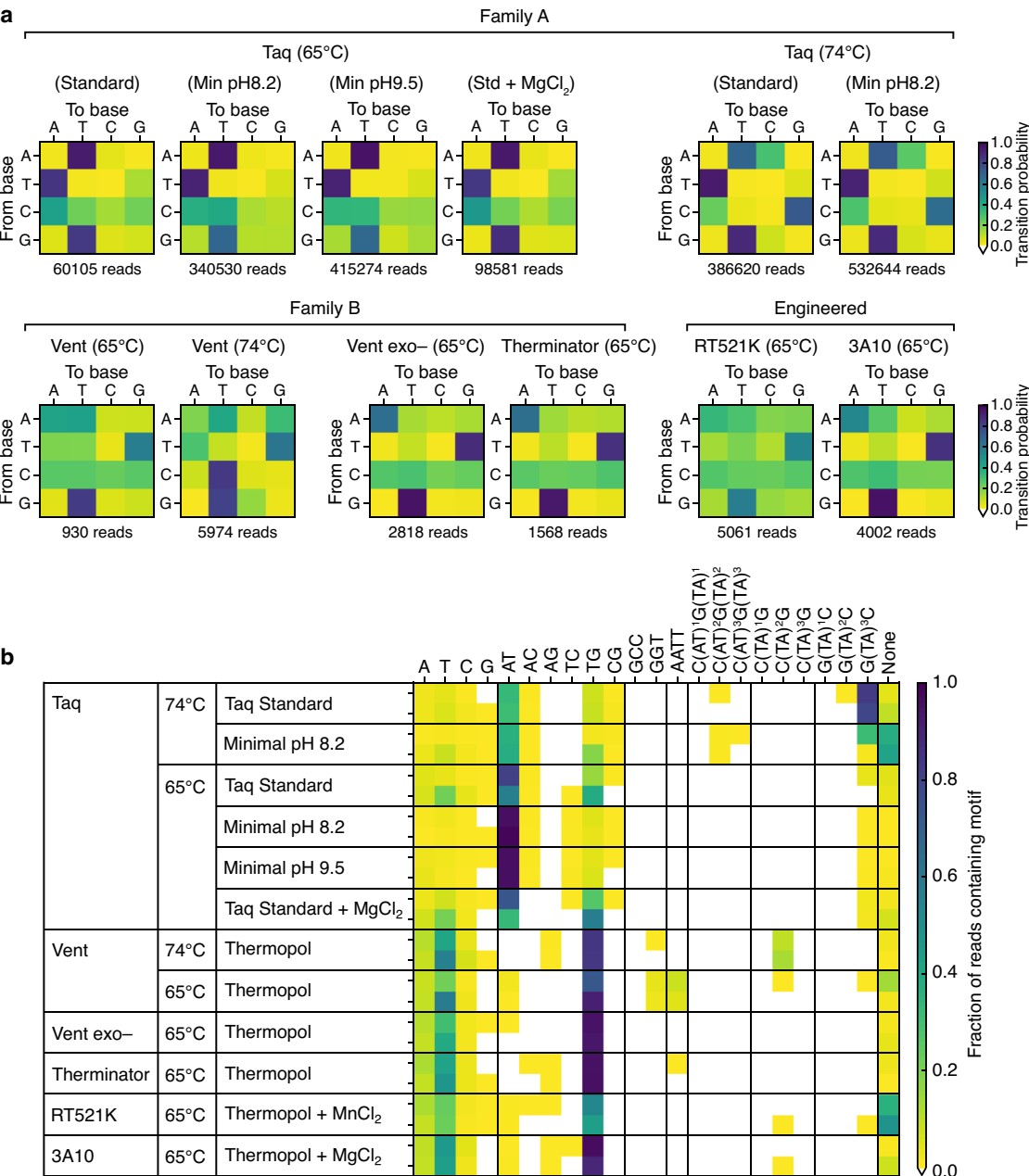

**Fig. 7 | Base transition probabilities and composition of repetitive sequence motifs in doodled DNA. a** Averaged base transition probabilities for Family A (Taq), Family B (Vent, Vent exo– and Therminator), and heavily engineered polymerases (RT521K: modified Tgo polymerase, and 3A10: chimera of Tth and Taq polymerase). Transition probability matrices shown for all reads >65 nt long. Number of reads for each polymerase shown below heat map. **b** Composition of repetitive sequence motifs in doodled DNA. Heatmap shows the fraction of reads containing repetitive sequence motifs (columns) for different DNA polymerases and reaction conditions (rows). Results generated by randomly sampling 600 reads (> 65 nt long) from the nanopore sequencing data and analyzing motif content of each separately. Data from the two experimental replicates is stacked with replicate 1 and 2 at the top and bottom, respectively. White areas of the heatmap denote motifs that were not present in any reads. To classify a read as containing a motif at least 7 consecutive repeats of 1–3 nt motifs, 6 consecutive repeats of 4 nt motifs, and 4 consecutive repeats of 6–8 nt motifs were required.

It is known that most DNAPs have a much higher propensity for base substitutions being transitions (A ↔ G or C ↔ T) as opposed to transversions (A ↔ T, C ↔ G, A ↔ C or T ↔ G). In particular, while A ↔ T and C ↔ G transitions in the doodled DNA products are prominent across all DNAPs (Fig. 7a), these types of transversion during base substitutions are typically rare being <10% and <5%, respectively, for many DNAPs[22]. This suggests that the structural constraints for doodling activity are very different to those affecting DNA copying fidelity and that these biases can be shaped through DNAP engineering.

## Analysis of repetitive sequence motifs

It is possible for a single DNA molecule to contain numerous different types of repetitive sequence throughout its length. These would not be captured by our previous autocorrelation analysis or through the calculation of transition frequencies (Figs. 2, 4 and 6). To better characterize the full range of repetitive motifs present in the sequencing data, we performed an analysis in which we searched for a set of common motifs found across all the polymerases and conditions assayed. We then calculated the fraction of reads in which these repetitive motifs were found (Fig. 7b). It should be noted that this

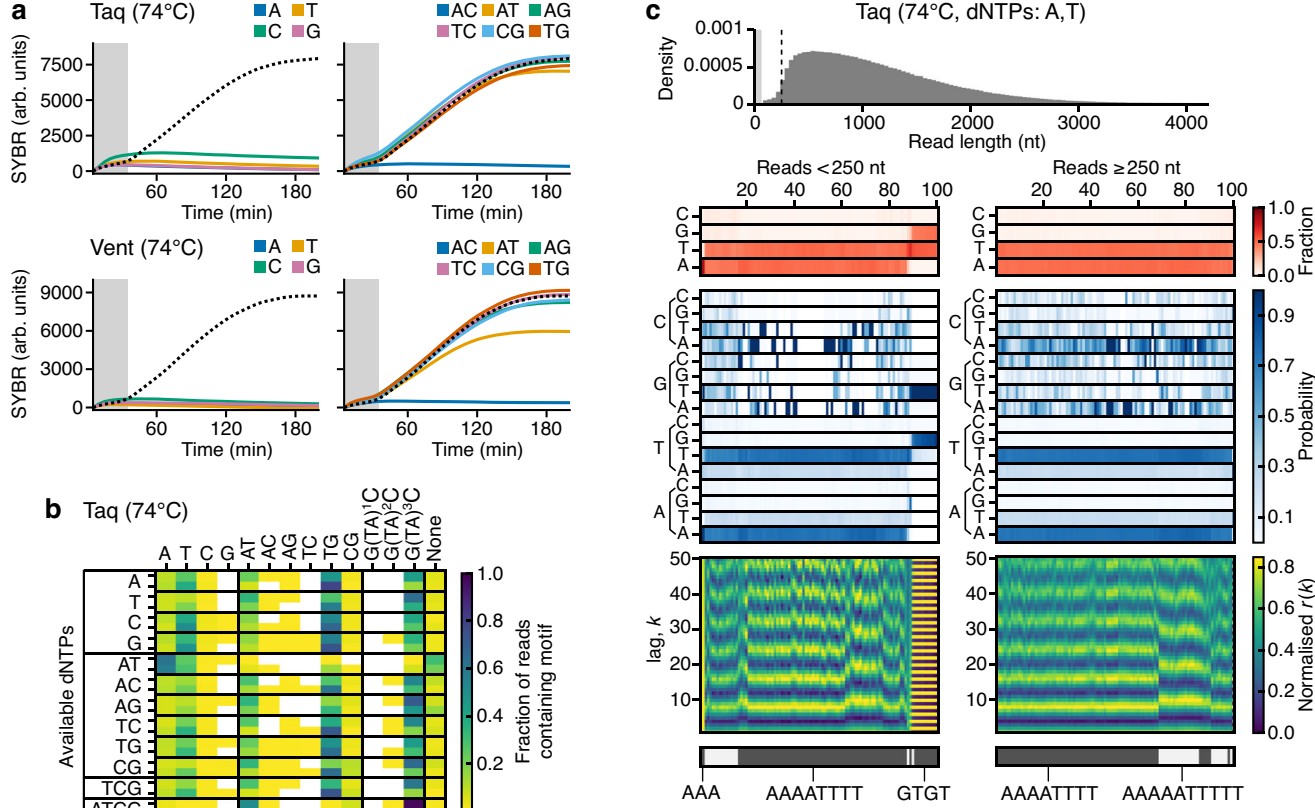

**Fig. 8 | Guiding sequence composition of doodled DNA using dNTP availability.**
**a** SYBR fluorescence for the Taq (top) and Vent (bottom) polymerases during the isothermal reactions in arbitrary units (arb. units). **b** Heatmap shows the fraction of reads containing repetitive sequence motifs (columns) for Taq polymerase in Taq Standard buffer at 74 °C for 16 h, with different mixtures of dNTPs (rows). Results generated by randomly sampling 600 reads (> 65 nt long) from the nanopore sequencing data and analyzing motif content of each separately. Data from the two experimental replicates is stacked with replicate 1 and 2 at the top and bottom, respectively. White areas of the heatmap denote motifs that were not present in any reads. To classify a read as containing a motif at least 7 consecutive repeats of 1–2 nt motifs, 6 consecutive repeats of 4 nt motifs, and 4 consecutive repeats of 6–8 nt motifs were required. **c** Sequence analysis for Taq at 74 °C with only adenine and

thymine present. Top histogram shows the sequence length distribution with the lightly shaded region denoting the 0–65 nt range and dashed line denoting the 250 nt read length. Below this, heatmaps show for a random subset of reads smaller and larger than 250 nt (left and right plots, respectively) the following information (top-bottom): 1. Sequence composition (red heatmap), 2. Probability of transitioning from one base to another (blue heatmap), 3. Autocorrelation analysis capturing the similarity of the sequence compared to itself after varying nucleotide shifts/lag $k$ (blue to yellow heatmap), and 4. the seven top clusters of reads (alternating light and dark gray) with key clusters having the sequence repeat they include below. Reads are displayed vertically and hierarchically clustered such that similar sequences are grouped.

analysis is more sensitive than the autocorrelation analysis to single base differences, with exact motif sequences matches required and a minimum number of repeats (at least 7 consecutive repeats of 1–3 nt motifs, 6 consecutive repeats of 4 nt motifs, and 4 consecutive repeats of 6–8 nt motifs).

From this analysis, we observed clear differences in the patterns of repetitive motif present for different polymerases, temperatures, and buffer compositions (Fig. 7b). One of the most striking features was the high frequency of AT repeats for Taq, and long T-homopolymers and TG repeats for the other polymerases, as well as the synthesis of reads with few repetitive motifs for RT521K and Taq at 74 °C with a minimal buffer at pH 8.2. More generally for both Taq and Vent, repeats of a longer length (> 4 nt) were more frequently seen at 74 °C compared to 65 °C, where shorter 1 or 2 nt long repeats dominated. Poly-G repeats were rare for all polymerases and CG repeats for all polymerases except Taq. However, the addition of $MgCl_2$ at 65 °C saw the complete loss of CG repeats for Taq. Very specific repeats also emerged for some of the polymerases under certain conditions: AATT and GGT repeats were uniquely seen for Vent at 65 °C, CTATAG repeats for Vent at 74 °C, and GTATAC and GTATATAC repeats for Taq at 74 °C in a Taq Standard buffer. Furthermore, there was a near complete breakdown of GTATATAC repeats for Taq at 74 °C when a minimal

buffer at pH 8.2 was used, highlighting how sensitive the synthesis processes are to environmental conditions.

## Guiding sequence composition using dNTP availability

Having verified that experimental conditions like temperature and buffer composition could influence doodling behavior, we next sought to determine whether availability of dNTPs could be used to guide the sequence of doodled DNA, providing a potential means by which some degree of control could be obtained over the sequence composition of the generated DNA fragments.

In the first instance, we carried out a set of additional real-time fluorescence experiments to determine whether the availability of only single types of dNTP or pairs of dNTP may influence the kinetics of doodling for the Taq and Vent polymerases (Fig. 8a). We found that for single types of dNTP virtually no doodling was observed for both polymerases. In contrast, when two different types of dNTP were present, consistent normalized DNA synthesis curves were obtained that closely matched reactions in which all dNTPs were present for both Taq and Vent (Fig. 8a). However, unlike reactions in which all dNTPs were present, most reactions with limited types of dNTPs synthesized much less DNA (< 4 ng/μL compared to >100 ng/μL) as compared with the situation when all dNTPs were present (Supplementary

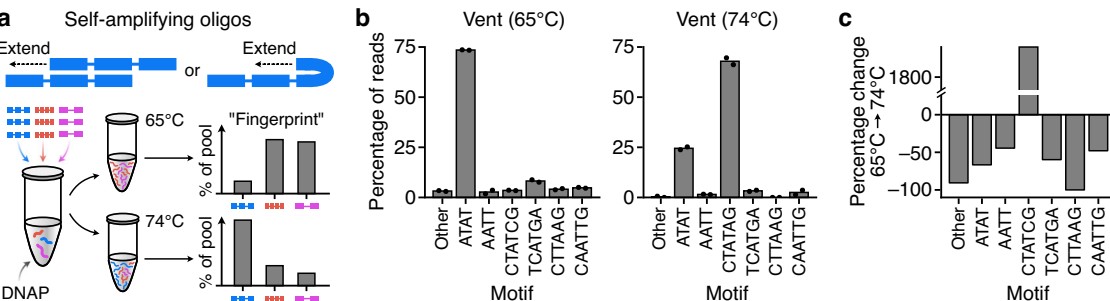

**Fig. 9 | Using seed oligos to guide doodling activity during isothermal reactions. a** Self-amplifying oligos are ssDNA molecules that contain repeats (colored blocks) that are able to form hairpins and base pair to themselves to allow for extension of the oligo by a DNA polymerase (DNAP). They can also base pair with other related molecules to enable extension. Mixtures of different designed self-amplifying oligos can be placed in a reaction with a DNAP and through the second stage of the doodling process (amplification) a unique "fingerprint" is produced in the sequences of the molecules present due to varying biases in the amplification process. **b** Average percentage of reads for seeded reactions containing 6 designed self-amplifying oligos and run using Vent at isothermal temperatures. Each reaction was performed in duplicate and data for each run is shown by black filled points. **c** Percentage change in the fraction of reads between 65 °C and 74 °C reactions and the dominant sequence motifs that were present in these.

Table 1). This is unsurprising given the biases previously observed in the ability of DNAPs to incorporate specific dNTPs into a DNA molecule when a base of a particular type is present at the 3' end.

To determine whether the limited dNTP availability affects the composition of the sequences produced, we performed nanopore sequencing on each reaction, as well as a further reaction containing three dNTP types (TCG). Analysis of the sequence motifs showed a bias for poly-T sequences and repeats of AT, TG and GTATATAC for all the single, double, and triple dNTP reactions, except for when only adenine and thymine were present (Fig. 8b). In that case, we found as expected that sequences were nearly entirely composed of those bases (87% of reads <250 nt and 99% of reads >250 nt). Interestingly, short and long reads were dominated by $A^4T^4$ repeats in addition to $A^5T^5$ repeats at a lower frequency (73% and 26%, respectively, across all reads >65 nt long) (Fig. 8c). While qualitative features of these repetitive sequence motifs were consistent across experimental replicates (Supplementary Fig. 15), much greater variation in the absolute fractions of specific motifs was seen compared to prior doodling reactions. Specifically, the $A^4T^4$ and $A^5T^5$ motifs that dominated both reactions saw drops of 16% and 10%, respectively across replicates. This suggests that these reactions may be more sensitive to minor differences in the setup of the reactions and external contaminants. Unlike all other reactions with limited combinations of dNTPs, where the majority of reads were <1000 nt (>80%), when only adenine and thymine were present >50% reads were long with a length >1000 nt. This suggests that these sequence motifs are more efficiently synthesized, likely due to their ability to self-replicate if the single-stranded DNA encoding them bends and base pairs with itself, allowing for standard extension and replication. Overall, these results demonstrate the potential to modify the composition of doodled DNA, in some cases by tuning dNTP availability, and may offer a means for the synthesis of long DNA fragments where the specific sequence is less consequential, but where statistical composition must be maintained.

It is unclear how DNA synthesis with bases not included in the reaction mix can take place, but could be due to contaminants in the individual dNTP mixes or polymerase preparations used (the commercial provider of the reagents used here guarantee ≥99% pure dNTPs determined by HPLC) as well as the high sensitivity of the nanopore sequencing. A less likely cause could also be the nanopore sequencing library preparation. However, such sequences have not been seen previously and do not match any of the barcode or adapter sequences used. If inclusion of these bases was due to contamination in the doodling reactions, their limited availability may account for the small amounts of DNA synthesized in these reactions (Supplementary Table 1).

Another possibility for the observed guanine and cytosine bases in the sequencing data is the high and sustained temperature of the doodling reactions, which could hydrolytic deamination of adenine to inosine. Given that the basecaller we use is not trained to detect inosine bases, if encountered, they could be miscalled as guanine or cytosine bases. To investigate this, we synthesized a pair of oligos (dIx3_F and dIx3_R; Supplementary Data 2) with complementary base-pairing flanks around a 3 bp central region that contained a repeat of inosine bases on the forward strand, and thymine on the reverse strand to mimic deamination of adenine bases (Supplementary Fig. 16a). These oligos were annealed to create dsDNA molecules and nanopore sequenced. Raw sequencing data was then basecalled, reads for each oligo separated, and the central region between the common flanking sequences extracted (**Methods**).

Reads of the forward oligo containing the inosine bases showed clear miscalling, with the majority of the central region given a G/C-based sequence, with CGC being the most common (32% in total) (Supplementary Fig. 16b). In contrast, the reverse strand saw accurate calling of the three thymine bases (72% in total) (Supplementary Fig. 16c). This suggests that some of the guanine and cytosine bases seen in our nanopore data may arise from miscalling of deaminated adenine bases. However, given the results for when only adenine and thymine are present, where >95% of bases across all reads matched these bases (Fig. 8b), it is unlikely that deamination is the major contributor to the unexpected guanine and cytosine bases seen in the sequencing data for the other doodling reactions with limited types of dNTP.

## Seeding reactions with self-replicating oligos to guide doodling activity

Having shown that doodling could be guided through changes in reaction conditions (e.g., temperature and available dNTPs), we finally wondered whether the amplification step could also be harnessed to sense changes in environmental conditions by seeding reactions with ssDNA oligos able to self-amplify. In particular, we took inspiration from differential sensing approaches that mimic our olfactory system and which have also been implemented in synthetic biology using de novo designed protein barrels[23].

To examine this, we developed a small panel of 6 ssDNA oligo sequence designs that contained repeat motifs allowing for self-amplification via either inter or intra DNA molecule binding (Fig. 9a; Methods). We considered repeat motifs of 2, 4 and 6 nt long and for the 6 nt motifs, elaborated on the CTATAG repeat seen in previous doodling reactions, altering the order of the T/A nucleotides in 4 different ways to create the motifs: CTATAG, TCATGA, CTTAAG,

CAATTG. These motifs, should have similar binding energies, but we hypothesized may have differing amplification efficiencies given only one of them had been seen during previous doodling reactions. Because the seed sequences would be provided to the reaction at the start, we also expected a stronger signal in the amplified pools and more consist and reproducible outcomes.

Isothermal doodling reactions were setup with equimolar amounts of each ssDNA seed oligo (0.2 μM total for all oligos combined) for the Vent DNAP. As a simple test, we exposed the reactions to 65 °C and 74 °C to see if a different fingerprint in the final DNA pool composition would be seen. As expected, we found major differences in the fraction that each oligo was present and saw high reproducibility across our replicates, with only very minor differences in the pool compositions (Fig. 9b). Interestingly, for the 6 nt motifs that all contained an identical nucleotide composition, there were substantial differences in the relative changes with >18-fold increase for the CTATAG motif, an ~50% reduction for the TCATGA and CAATTG motifs, and a near 100% reduction in the CTTAAG motif (Fig. 9c). This suggests that the amplification process is highly sensitive to minor differences in the specific ordering of nucleotides of the seed oligos, a feature that could potentially be exploited to aid discrimination of other environmental factors that might affect aspects of DNA folding and binding to other DNA molecules and the DNAP.

These results demonstrate the ability to exploit the differential amplification of ssDNA molecules during isothermal doodling reactions for encoding features of the reaction environment into the composition of the pool of DNA molecules that are synthesized, suggesting yet a further application of this process beyond de novo DNA synthesis.

## Discussion

In this study we used long-read sequencing, real-time fluorescence assays, and atomic force microscopy to perform the most detailed analysis to date of untemplated DNA synthesis. This allowed us to obtained further insights into the sequence composition, structure and kinetics of the DNA fragment pools produced by natural and engineered DNAPs. Prior studies have characterized just a small number of separately cloned DNA sequences[10], making it impossible to understand the full diversity of the sequences produced by untemplated synthesis activity, which hampers the identification of rare sequences that may be of significance. Our results suggest that there may be distinct signatures of doodling activity resulting from biases in some of the base transitions and repetitive motifs. Further analysis of the occurrence of such signatures across genomes spanning all phylogeny could provide insights into a putative ongoing function of untemplated DNA synthesis in genome evolution. Furthermore, building on prior work[2,9,10,13], we have shown that the sequence signatures observed are DNAP-dependent and vary according to environmental conditions and reaction buffer composition. Tuning these parameters suggests a potential way in which the statistical features of doodled DNA sequences may be systematically manipulated and programmed. This has the potential to form the basis of a DNA synthesis method that could complement current chemical and enzymatic oligonucleotide assembly methods.

Some efforts to exploit untemplated DNA synthesis by other enzymes have been made. Systems have been built that use the inherent sensitivity of terminal deoxynucleotidyl transferases (TdTs) to physiologically relevant signals like $Co^{2+}$, $Ca^{2+}$, $Zn^{2+}$ and temperature to affect their template-independent extension as a means to encode environmental changes into DNA with a temporal resolution of minutes[24]. Beyond direct synthesis, the evolution of biased DNA pools when a DNAP is present has also recently been investigated[25]. Starting from a pool of purely adenine and thymine or guanine and cytosine DNA oligonucleotides (12 nt long) with a bias of purine or pyrimidine nucleotides, a *Bacillus stearothermophilus* strand displacing polymerase (Bst) was added and evolution of the pool monitored. It was found that initial biases in the pool disappeared as the oligonucleotides were extended and highly repetitive dimer and trimer motifs in rapidly extended sequences emerged. Analysis of these sequences revealed the importance of sequences supporting hairpin formation and reverse-complementary 3-mer periodic motifs. These findings provide some insights into the biophysical basis of the biases observed and may offer a way to better predict repetitive motifs that are likely to efficiently replicate if generated via a doodling reaction.

A deeper mechanistic understanding of template-free DNA synthesis may also have implications beyond de novo DNA construction. Many in vitro diagnostic workflows involve sequencing low-copy samples, where positive specimens may contain only a few molecules of target DNA and negative controls contain none. Under such conditions, even modest levels of untemplated synthesis could contribute additional background signal, analogous to the ab initio activity reported for multiple DNA polymerases[5] and the emergence of non-specific, repetitive products observed during polymerase-driven evolution of short oligonucleotide pools[25]. Although the extent to which such processes influence current diagnostic assays remains unclear, our results show that doodling produces characteristic sequence signatures–including biased transition probabilities and highly repetitive motifs – that could, in principle, be detected and filtered bioinformatically. Given the central role of isothermal and whole-genome amplification in many in vitro diagnostic pipelines, improved understanding of how to suppress, modulate, or computationally remove such background products may help enhance sensitivity and reduce false-positive risk in ultra-low-input sequencing applications.

The clear signatures seen in doodled DNA opens interesting questions regarding whether these types of DNA molecule are synthesized in vivo and their potential role during evolution. A quick BLAST search for the 30-mer sequence (CTATAG)[5], a long repeat of one of the motifs we find, across the NCBI core nucleotide database yields large numbers of hits from across the tree of life, with many being perfect matches. Such sequences are unlikely to occur by random chance ($p = 8.67 \times 10^{-19}$, assuming all nucleotides are equally probable). However, it remains to be seen if these patterns are the result of prior doodling activity or if they arose through other means. An intriguing future direction for this work would be to perform a detailed analysis of common doodled motifs throughout known genome sequence space, to assess whether doodling might have played a role in fueling early evolution (e.g., through comparisons of motifs in genes of early origins), allowing doodled products to act as a feedstock for natural selection to act upon. Any such analysis would need to employ careful statistical analyses to ensure other evolutionary processes did not confound the results. It may also be more productive to consider using bottom-up synthetic biology approaches in which systems can be more thoroughly observed and precisely controlled over time.

One major challenge we faced when working with pools of doodled DNA was the extreme length of some fragments. While the read length distributions we report give some idea of differences in fragment lengths between experiments, it is likely that very long DNA fragments are not captured during sequence analysis due to fragmentation by shearing during preparation of the sequencing libraries. We have used barcoding to enable the multiplexing of samples, but this introduces additional liquid handling steps that may have increased the extent and frequency of DNA shearing. We did carry out additional sequencing experiments where the barcoding stage was omitted as well as gentler processing of the DNA, and found that there was a small shift in the read length distributions towards longer reads, and did capture some reads >23 kb (maximum lengths of 29,009 nt and 85,364 nt for Taq and Vent, respectively). The distributions presented here are therefore likely to represent the lower boundary of the actual DNA fragment lengths generated by doodling. Developing new

library preparation methods that mitigate shearing and performing deeper sequencing would be an interesting future direction of this work, allowing us to better understand the full spectrum of molecules produced[26,27].

The ability to use doodling for the production of long DNA fragments (potentially >100 kb) differentiates it from existing technologies that are typically based on relatively slow cyclic processes to enable the syntheses of specific DNA sequences[28]. However, our results show that while the native doodling capabilities of DNAPs vary in response to specific environmental parameters, these changes are limited and cannot yet be adequately controlled or manipulated to support the synthesis of specific DNA sequences. Developments in long-read sequencing through the engineering of polymerases and protein nanopores and their interfacing with electrical and photonic systems have demonstrated the leaps that can be made to rapidly harness and repurpose biological functions[29–34]. Therefore, an interesting future direction would be to explore whether the interfacing of DNAPs with electronic systems might facilitate the rapid modulation of environmental or catalytic biases in the DNAP and greater control over the doodling process and composition of the DNA pools synthesized. When combined with emerging capabilities in AI-powered protein design[35], improvements in the fidelity of synthesized sequences by engineered DNAPs could be closer than we think.

The vertebrate immune system uses TdTs to add nucleotides without any template in order to increase antigen receptor diversity[36] and was previously considered the only enzyme able to generate DNA without a template. Enzymatic DNA synthesis using TdTs has spawned several successful companies such as Molecular Assemblies, DNA Script, and Ansa Biotechnologies[37] who have developed specialized protocols, 3′-protection groups[38–40], and re-engineered TdTs with more rapid nucleotide addition (Patent WO/2019/135007), modified catalytic activity[41], and with dNTPs tethered to the enzyme[28]. These advances have helped mitigate some of the limitations of TdTs, including the bias towards the addition of particular nucleotides[42]. Development of DNAP-based enzymatic DNA synthesis may similarly improve the control of untemplated DNA synthesis and provide further benefits. For example, TdT approaches typically struggle to produce highly repetitive and structurally prone sequences that are known to be important structurally in virus genomes (e.g., in terminal repeats) and which are present throughout the human genome (e.g., Alu repeats are 11% of the human genome and modify gene expression). The demonstrated ability for doodling by DNAPs to effectively synthesize repetitive sequences would fills a gap in current TdT capabilities and could help us to further understand the effects of repeat sequences in nature and engineered biological systems.

Beyond sequence complexity, the yield, purity and length of DNA synthesized by any methodology depends on the efficiency of each cycle of nucleotide addition. Despite the excellent reported efficiency from DNA Script of 99.7%[43], mathematically, the yield of products of 1 kb in length is only ~5%, and even 99.9% efficiency yields <37% of 1 kb length products[37], not taking into consideration the error rate. Longer DNA constructs have been made using DNA assembly methods, namely a 10 kb construct from Ribbon Biolabs using Gibson assembly (Patent WO/2019/073072) and whole genomes using yeast-based DNA assembly[44,45]. However, assembly of DNA fragments can be laborious and expensive so the development of technologies able to write longer and more accurate DNA constructs will be vital in advancing synthetic genomics. Furthermore, while the fidelity of doodling at present means it is impossible to synthesize long and defined sequences, the ability of DNAP-based methods to both carry out templated and untemplated DNA synthesis, could open up new avenues for DNA production that exploit de novo synthesis and amplification concurrently – features that no existing DNA synthesis technologies combine.

Although producing precisely defined sequences is likely to be a challenge in the short, or even medium term, if sufficient improvements to the control of the doodling process can be made, it may one day be possible to rapidly and efficiently produce long ssDNA fragments of defined composition. This would have significant implications for a broad range of biotechnological applications. In the near term, the ability to accurately synthesize long DNA molecules rapidly, and at low cost, will have substantive utility in accelerating large-scale genetic circuit construction[46] and could act as a interesting substrate for material sciences, e.g., DNA-based "glues"[28,47,48]. Furthermore, the ability for the sequences generated to dynamically vary in response to specific environmental conditions provides a means for recording events as sequences in stable DNA molecules (e.g., similar to Bhan et al.[24].).

Over the longer term, this work builds on ongoing efforts to establish a foundation for exploring novel, biologically inspired approaches to DNA synthesis. Compared to existing approaches (such as DNA synthesis using phosphoramidite chemistry) these would have a less significant environmental impact, and would be faster, more efficient, and have the potential to generate DNA sequences of virtually any length[28,49]. This type of technology is exactly what will be required for biological engineering as it transitions from single-gene interventions to system-level genome design and construction, and is broadly utilized as a predictive engineering material[50,51]. This work supports these efforts and further demonstrates the value of long-read sequencing and other physical measurement techniques (i.e., AFM) when attempting to unpick complex genetic processes[33,52,53].

## Methods

### Standard in vitro reactions

All doodling reactions were performed in thin wall PCR tubes or 96-well PCR plates, using reaction volumes of 100 μL with 200 μM of dNTPs (New England Biolabs, N0447S). Reactions were prepared on ice in a PCR hood before being transferred to a preheated thermocycler (Bio-Rad, C1000). Nuclease-free water (Omega Biotek, PD092) was used to prepare all reactions. Reactions were run for 16 h at 64 °C or 75 °C, unless stated otherwise. Taq reactions were performed in a 1X Taq Standard Buffer (New England Biolabs, B9014) containing 10 mM Tris-HCl, 50 mM KCl, 1.5 mM $MgCl_2$, and 5 units of polymerase (New England Biolabs M0273). Vent reactions were performed in 1X Thermopol buffer (New England Biolabs, B9004) containing 20 mM Tris-HCl, 10 mM $(NH_4)_2SO_4$, 10 mM KCl, 2 mM $MgSO_4$, 0.1% Triton X-100, and 2 units of polymerase (New England Biolabs M0254). Vent exo− (New England Biolabs, M0257) and Therminator (New England Biolabs, M0261) polymerase reactions were set up similarly to those for Vent. RT521K polymerase reactions were performed in 1X Thermopol buffer supplemented with 25 Mm $MnCl_2$. 3A10 polymerase reactions were performed in 1X Taq standard buffer supplemented with 25 mM $MgCl_2$. For RT521K and 3A19 polymerase reaction, 1 μL (5 U/μL) of polymerase was used. 1X minimal buffer was formulated as 10 mM Tris-HCl, 50 mM KCl, 2.5 mM $MgCl_2$ and adjusted to pH 8.2 at 25 °C or pH 9.5 at 25 °C.

### In vitro reactions with temperature switching

Temperature switching reactions were performed in thin-wall PCR tubes, using reaction volumes of 100 μL with 200 μM of dNTPs (New England Biolabs, N0447S). Reactions were prepared on ice in a PCR hood before being transferred to a preheated thermocycler (Bio-Rad, C1000). Nuclease-free water (Omega Biotek, PD092) was used to prepare all reactions. Duplicate reactions were performed over a total of 6 h, consisting of 9 cycles of 64 °C for 20 min followed by 75 °C for 20 min. Taq reactions were performed in a 1X Taq Standard Buffer (New England Biolabs, B9014), and 5 units of polymerase (New England Biolabs, M0273). Vent reactions were performed in 1X Thermopol

buffer (New England Biolabs, B9004), and 2 units of polymerase (New England Biolabs, M0254). Prior to sequencing, PCR products were purified using Beckman Colter AMPure XP beads (Fisher scientific, A63880) at a bead-to-sample ratio of 1.8:1.

### In vitro reactions with seed oligos

Isothermal amplification reactions were performed in thin-walled PCR tubes with 100 μL total reaction volumes. Reaction mixtures contained 200 μM dNTPs (New England Biolabs, N0447S), 0.2 μM of a single seed oligo (seed_ATAT_5, seed_AATT_5, seed_CTATAG_3, seed_TCATGA_3, seed_CTTAAG_3, seed_CAATTG_3; Supplementary Data 2), 1X ThermoPol buffer (New England Biolabs, B9004), and 2 μL of Vent DNA polymerase (New England Biolabs, M0254). Nuclease-free water (Omega Biotek, PD092) was used in all reactions. Reactions were assembled on ice in a PCR hood and incubated isothermally at 65 °C and 74 °C for 16 h. A no-seed oligo control was included. For pooled seed oligo reactions, six seed oligos were combined to a final concentration of 0.6 μM (0.1 μM each). Reactions were performed in 100 μL volumes containing 200 μM dNTPs (New England Biolabs, N0447S), 1X ThermoPol buffer (New England Biolabs, B9004), and 2 μL of Vent DNA polymerase (New England Biolabs, M0254). Nuclease-free water (Omega Biotek, PD092) was added to a final volume of 100 μL. Reactions were incubated isothermally at 65 °C and 74 °C for 16 h and were performed in duplicate. Prior to sequencing, PCR products were purified using Beckman Colter AMPure XP beads (Fisher scientific, A63880) at a bead-to-sample ratio of 1.8:1.

### Gel electrophoresis

Gel electrophoresis was performed for DNA length quantification of reaction products. Gels were prepared using 1% agarose in 30–50 mL of 1X TAE running buffer (40 mM Tris-acetate, 1 mM EDTA). Gels were stained by adding 1X nucleic acid stain Gel Green (Biotium, 41005) to the gel solution prior to casting. 50 μL of sample was loaded into each lane of the gel, using 10 μL purple loading dye (New England Biolabs, B7025S). The DNA 1 kb+ ladder (New England Biolabs, N3200) was used as a reference. Gels were run at 80 V for 30–60 min (Bio-Rad, PowerPac Basic).

### Nanopore DNA sequencing

For most samples, DNA was prepared for nanopore sequencing following the ligation sequencing of amplicons protocol using Oxford Nanopore Technologies (ONT) kit SQK-LSK110 directly, or the native barcoding protocol using ONT kits SQK-LSK110 and EXP-NBD104. DNA quantification was performed using a Qubit dsDNA broad range DNA quantification kit and a Qubit 4 fluorometer (Invitrogen, Q33238). For barcoding, equal volumes of each barcoded sample were pooled and 6–60 ng of DNA was loaded into each flow cell (ONT, FLO-MIN106). Sequencing was performed using a MinION Mk1B device and the MinKNOW software version 21.11.7. DNA samples for temperature switching and seed oligo experiments were prepared for nanopore sequencing using the Native Barcoding Kit 24 V14 (ONT, SQK-NBD114.24) and loaded onto FLO-MIN114 flow cells. For barcoding, equal volumes of each barcoded sample were pooled and 8–112 ng of DNA was loaded onto the flow cell. For seed oligo experiments, individual seed oligos and pooled oligos, incubated at either 65 °C or 74 °C, were loaded onto the flow cell at a final pooled DNA concentration of 566 ng. The sequencing library preparation for the high molecular weight samples used an adapted protocol for the Ligation Sequencing DNA V14 kit (ONT, SQK-LSK114), with libraries loaded onto FLO-MIN114 flow cells. Specifically, only a single sample was sequenced per flow cell to reduce the number of clean up steps required before sequencing (i.e., clean up steps after barcode ligations), and all steps were carried out using wide bore pipette tips and performed very slowly to minimize the chance of DNA shearing throughout.

### Real-time tracking of DNA synthesis using fluorescence

A real-time PCR machine was used to monitor using fluorescence the synthesis of doodled DNA products during a reaction. For the Taq polymerase, 100 μL reactions were prepared on ice in 96-well plates containing 84 μL nuclease free water (Omega Biotek, PD092), 10 μL of 10X Taq standard buffer (New England Biolabs, B9014), 2 μL of 10 mM dNTPs (complete mixtures of all dNTPs, New England Biolabs, N0447S; or combinations of single types of dNTP supplied as a separate set, New England Biolabs, N0446S), 1 μL Taq polymerase (2 units final concentration), 2 μL ROX solution (ThermoFisher Scientific, R1371), and 1 μL SYBR Green dye (ThermoFisher Scientific, S7563). For the Vent polymerase, identical reactions were prepared, however, the 10X Taq standard buffer was replaced with a 10X Thermopol buffer (New England Biolabs, B9004). Assembled reactions were then run in a preheated qPCR machine (Agilent Technologies, Stratagene MXP3005P) with a fixed temperature of 74 °C and a cycle time of 5 min for a total of 16 hours. Measurements of fluorescence (SYBR Green and ROX) were taken each cycle.

### Atomic force microscopy

AFM was performed on Taq DNA samples generated using 100 μL reactions. Reactions were assembled in batch to minimize pipetting error and contained 1X Standard Taq Buffer (New England Biolabs, B9014), 200 μM dNTPs (New England Biolabs, N0447S), 5 units of Taq polymerase (New England Biolabs, M0273), and nuclease-free water (Invitrogen). Reactions were incubated at 74 °C for 16 h and performed in triplicate. DNA was subsequently purified using AMPure XP beads (Beckman Colter, A63880) at a 1.8:1 bead-to-sample ratio and eluted in 40 μL nuclease-free water. Samples were diluted in Tris-MgCl₂ deposition buffer (10 mM Tris-HCl, 5 mM MgCl2, pH 7.6), freshly prepared and heated to 80 °C for 1 h prior to use, to a final concentration of 0.02 μg/mL for AFM imaging. To enable high-resolution imaging of individual DNA molecules, the samples were deposited onto an atomically flat freshly cleaved mica substrate. This surface facilitates optimal contrast and resolution for visualizing double-stranded DNA, which has a diameter of ~2 nm. A 1 μL DNA sample was applied to the freshly-cleaved mica surface and incubated for 1.5 min, followed by three washes with 200 μL Milli-Q water. The samples were then dried using a gentle stream of dry air to remove any water droplets. Mica disks were then baked at 120 °C for 20 min, cooled to 40 °C (covered dry bath followed by covered bead bath). Finally, the prepared mica substrates were affixed to SEM stubs and imaged using a high-speed atomic force microscope (HS-AFM).

The AFM used for this study was a custom contact-mode HS-AFM developed at the University of Bristol, UK[54]. It is able to generate two 1-megapixel topographic maps per second with a height resolution of ±15 picometers and has an adjustable scan size of 500 × 500 nm to 30 μm × 15 μm per image. Previous versions of this instrument have demonstrated the feasibility of using contact-mode HS-AFMs as genomic diagnostic tools[55]. These studies have shown that the high-spatial resolution and rapid imaging capabilities can be used to generate large ( > 1.5 million molecule) data sets in practical timescales and at competitive price points per test[56]. Here, we used the high throughput of the HS-AFM to survey the high molecular weight samples of doodled DNA and measure key characteristics such as the backbone length and number of branches of over 700 individual DNA molecules.

For analysis, it was necessary to adapt existing image analysis algorithms for DNA detection and measurement because many of the doodled DNA molecules featured one or more branches. Typical DNA analysis routines cannot distinguish the difference between a single branching DNA molecule and multiple overlaid molecules. For this study, a semi-automated approach was implemented. In brief, each raw AFM image was flattened using a line-by-line 3rd order polynomial, and a threshold used to identify DNA molecules. After

this step the operator inspected each detected DNA molecule and in the case of any strands with branches, they identified the branching locations by manually selecting the junctions with the mouse pointer. The backbone and any branches were then traced using a spline fit to the skeletonized threshold image. This enabled each molecule to be defined not just in terms of its backbone length but also by the number and length of any branches from this backbone.

## Sequencing data analysis

Raw sequencing data in FAST5 format was basecalled using guppy version 6.3.8 and the configuration file 'dna_r9.4.1_450bps_hac.cfg'. The resultant FASTQ files were then cleaned by trimming any adapter or barcode sequences found at the start or end of each read and reads with barcodes found internally were removed. Unless otherwise stated, we also filtered reads based on length >65 nt and Q-score ≥13 to enable more accurate inference of statistical and repetitive features of the sequences. Autocorrelation analyses were performed by calculating a discrete and normalized string-based autocorrelation function $r(k)$, where $k$ is the lag (shift in sequence). We assume that only perfectly matching elements (bases) are included in the calculation with the final value normalized by sequence length to account for the finite nature of the input sequence. This was implemented as a custom function in Julia.

## Analysis of contaminant sequences

To analyze the source of contaminant sequences in the temperature switching experiments, we performed a detailed annotation of a random selection of reads using the pLannotate tool[57]. Selected sequences were manually extracted from the processed FASTQ file for replicate 1 of the temperature switching experiment and annotations generated using the pLannotate web interface. CSV files containing the annotations for each read were downloaded and then compiled into a single spreadsheet (Supplementary Data S1).

## Analysis of E. coli genome for repetitive sequence motifs

Analysis of the *E. coli* genome was performed using the complete sequence of the BL21(DE3) strain (GenBank: CP001509.3). Julia scripts were used to search for occurrences of repetitive sequence motifs that were able to self-replicate and which were seen in the doodling reactions. The proportion of the genome encoding them calculated as $(n \times m)/g$, where $n$ and $m$ are the number of occurrences and length of the motif, respectively, and $g$ is the total length of the *E. coli* BL21(DE3) genome sequence (4,558,953 bp). We specifically searched for the following motifs: $(CTATAG)^2$, $(CTATATAG)^2$, $(CATATG)^2$, $(CATATATG)^2$, $(GTATAC)^2$, $(GTATATAC)^2$, $(GATATC)^2$ and $(GATATATC)^2$. Only perfect matches were considered.

## General computational analysis tools

All analysis scripts were developed and run using Julia version 1.10.2 and the following packages: BioAlignments version 3.1.0, BioSequences version 3.1.6, Clustering version 0.15.7, CSV version 0.10.14, DataFrames version 1.6.1, Dates version 1.10.0, Distances version 0.10.11, FASTX version 2.1.4, Glob version 1.3.1, LinearAlgebra version 1.10.0, Random version 1.10.0, and Statistics version 1.10.0. Plots were generated using Julia and the CairoMakie version 0.12.3 package and final figures composited in Affinity Designer version 2.5.3.

## Statistics and reproducibility

Sample sizes were chosen to match typical practices in the field (i.e., biological duplicates for DNA sequencing data). No data were excluded from the analyses. The experiments were not randomized. The Investigators were not blinded to allocation during experiments and outcome assessment.

## Reporting summary

Further information on research design is available in the Nature Portfolio Reporting Summary linked to this article.

## Data availability

The DNA sequencing data generated in this study have been deposited in the European Nucleotide Archive (ENA) under accession code PRJEB107084 (https://www.ebi.ac.uk/ena/browser/view/PRJEB107084). In addition, sequencing data (raw and processed), real-time PCR data and AFM images are available from Zenodo at: https://doi.org/10.5281/zenodo.17956595. Source data are provided with this paper.

## Code availability

Analysis scripts are available from Zenodo at: https://doi.org/10.5281/zenodo.17956595.

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

## Acknowledgements

We would like to thank the anonymous reviewers for their valuable comments and suggestions that have significantly strengthened this work. This research was supported by Replay Holdings Inc., a Royal Society University Research Fellowship grant URF\R\221008 (T.E.G.), a Turing Fellowship from The Alan Turing Institute under the Engineering and Physical Sciences Research Council (EPSRC) grant EP/N510129/1 (T.E.G.), BrisEngBio, a UKRI Engineering Biology Transition Award grant BB/W013959/1 (T.E.G.), and by the Medical Research Council (MRC) as part of United Kingdom Research and Innovation (UKRI) MRC program grant MC_U105178804 (P.H.) We would also like to thank the Replay Genome Writing Team for insightful discussions and feedback on the research.

## Author contributions

A.W., A.H., T.E.G., L.M.O. and G.L. conceived the project. T.E.G., S.D.C., A.W. and G.L. designed the experiments. P.H. provided the 3A10 and RT521K engineered DNA polymerases for testing. S.D.C., T.C.T.I. and T.E.G. performed the sequencing experiments. S.D.C. performed the real-time fluorescence assays. L.P. performed and analyzed data from the AFM experiments. T.C.T.I. supported the AFM experiments. T.E.G. and S.D.C. carried out all the analyses of the sequencing data with input from G.L., B.T.R. and I.D.W.S. T.E.G. supervised all experiments and wrote the initial manuscript. All authors contributed to the interpretation of the results and final editing of the manuscript.

## Competing interests

A.W., G.L., B.T.R. and L.M.O. have been employees of Replay Holdings Inc. T.E.G., I.D.W.S., A.H. and P.H. have consulted for Replay Holdings Inc. The remaining authors declare no competing interests.
