## [Transparent Peer Review file · Nature Communications]

Analysis and control of untemplated DNA polymerase activity for guided synthesis of kilobase-scale DNA sequences

Corresponding Author: Professor Thomas Gorochofski

Version 0:

Reviewer comments:

Reviewer #1

(Remarks to the Author)

This manuscript from Castle et al looks at the untemplated DNA synthesis activity of several DNA polymerases. In particular, they use long read synthesis to analyze the sequence composition resulting from the untemplated activity of the polymerases under a few temperature and buffer conditions.

Although the topic is interesting, the data resulting from the characterization is potentially useful, and the experimental approach is reasonable, I feel it is not sufficiently interesting and not sufficiently useful (no major insights, nor realistic potential applications), and the depth of analysis is insufficiently thorough to merit publication in Nature Communications. One general concern with this type of experimental approach is that the data can be easily skewed by contamination. The authors acknowledge likely contamination of the individual NTP mixes with other NTPs and it seems plausible that, in the absence of a careful analytical characterization of the purity of the NTPs and other ingredients the results are significantly affected by contamination, skewed ratios of NTPs in the mixes, or even contamination with DNA. The authors state that the patterns were "robust and reproducible across replicates", however the replicate data in the supplementary material show readily noticeable differences. It is not clear how to interpret such differences, but in the absence of a well-defined metric, it seems that such a claim of reproducibility cannot be easily made.

I do believe, a priori, that different polymerases and different environmental conditions can result in reasonably well defined patterns of nucleotide incorporation, but this is neither surprising nor necessarily as useful as the authors suggest. The authors suggest or imply that it might be possible to control the "doodling" activity to the point of reaching something approaching sequence-controlled synthesis. In my opinion this is farfetched, particularly given the existence of polymerases like TdT, which are more efficient for untemplated synthesis and are approaching full controllability. Even the title mentions "control of untemplated DNA polymerase activity" but this is far from what is actually shown.

Also missing from the manuscript is an explanation and detailed analysis/interpretation of the histograms obtained from the sequencing.

(Remarks on code availability)

Reviewer #2

(Remarks to the Author)

The manuscript from Castle et al describes analysis of untemplated DNA synthesis by DNA polymerases. This is an interesting but poorly understood area, and I think this study is very interesting and presents a new level of insight into the DNA made by this unusual mechanism. I think there are a couple points that should be addressed and updated, but the paper is mostly suitable for publication and adds a significant contribution to the understanding of untemplated polymerase activity.

1) The presence of G and C bases in the reactions containing only dA/TTP is puzzling, which the authors admit. The suggestion of dNTP contamination seems pretty unlikely, as the G/C-containing sequences are pretty prevalent and a significant amount of the bases would have to be present. The reactions are heated at high temperature for very long times, could some deamination of dA to dI give a signal read as G or C? Nucleotides are easily analyzed by mass spec and the vendor here likely could share any contaminant levels. I feel this is a mysterious result worth a bit more explanation.

- 2) The authors acknowledge that the read length distributions might be biased to shorter products, but I think this is more significant than discussed. From the agarose gels in the paper it would seem the DNA products are overwhelmingly many kb in length, but the sequencing reads are predominantly very short. Could some of the longer DNAs be branched? Single-stranded? Would a HMW-focused library prep give reads more in line with the nature of the products? My concern is that the reads aren't reflective of the real nature of the DNA products and I think this could be examined more.
- 3) The Family A and B polymerases used here give different transition frequencies, are these similar to published analysis of the polymerases error spectra when fidelity and misincorporation are being examined?
- 4) This is perhaps a bit pedantic, but I would suggest the authors not state they're using "real-time PCR" to measure the production of the DNA. They're doing isothermal reactions monitored by real-time fluorescence, no PCR is occurring at all just using a "qPCR machine" as a fluorimeter. I think these descriptions should be adjusted accordingly. Although it would be interesting to see what DNA products would be produced if these reactions were thermocycled, the introduction of denaturing steps might give reason to annealing of "primer" strands and different DNA.

(Remarks on code availability)

Version 1:

Reviewer comments:

Reviewer #1

(Remarks to the Author)

The revised manuscript has been significantly extended with the addition of new experimental results as well as extensive revisions to the text, in both cases to address concerns from the reviewers. In my opinion, the revisions have significantly improved the manuscript, particularly by addressing the possibility that some of the experimental results may be due to DNA contamination. Given these improvements, I recommend that the manuscript be published in Nature Communications with minor changes. In particular:

1. In my view, the value of the "doodling" to DNA synthesis is oversold. For example, the statement (line 496) to the effect that it could be disruptive to current chemical or TdT-based synthesis cannot be justified by the results presented. I suggest that this type of statement, here and elsewhere, be toned down a couple of notches, as there is almost no conceivable path that would allow doodling to reach even the level of control available via current TdT variants or chemical synthesis.

2. I think the interest in the author's results will be more in the direction of potential signatures of doodling in natural systems. However, even here, I think the authors need to be more careful with unsupported statements. The statement (lines 530 to 539) regarding hits in nucleotide databases of e.g. CTATAG, needs to be either removed or supported with a statistical analysis. Any 6mer is going to be present in very large numbers in any large genome, and even if the doodling mechanism had some role in early evolution, it seems likely that a few billion years of evolution would have any evidence of this.

(Remarks on code availability)

Reviewer #2

(Remarks to the Author)

The authors have, in my view, done an excellent job responding to reviewer comments and have added a significant amount of new data. These additions result in a more complete and clearer investigation of the interesting activities of the DNA polymerases presented. The AFM data in particular helps clear up one of the biggest questions I had from the first version. My only minor issue is that the branched DNAs were produced by Taq, which given its 5'-3' exonuclease activity does not have strand displacement ability and generally does not make branched DNA like you get in RCA/MDA/etc. So these branches must not be 5' extensions but rather something else? More a note of curiosity than something the authors need to address.

Overall I think this is now an impressive collection of data that explores the findings very thoroughly and with more than enough discussion of the various hypotheses and potential applications of the phenomenon. In my opinion this manuscript is suitable for publication as is and serves as an important contribution to this field.

(Remarks on code availability)

Response to reviewers (NCOMMS-24-60521)

Find below our responses (in blue) to all the reviewers' comments. We thank them for their valuable feedback and believe that our updates to the manuscript address all their concerns. We have also carefully reviewed the entire manuscript and made minor changes throughout to improve readability. In the revised manuscript, sentences containing edits have been entirely highlighted in red to aid reviewing.

Reviewer #1:

This manuscript from Castle et al looks at the untemplated DNA synthesis activity of several DNA polymerases. In particular, they use long read synthesis to analyze the sequence composition resulting from the untemplated activity of the polymerases under a few temperature and buffer conditions.

Although the topic is interesting, the data resulting from the characterization is potentially useful, and the experimental approach is reasonable, I feel it is not sufficiently interesting and not sufficiently useful (no major insights, nor realistic potential applications), and the depth of analysis is insufficiently thorough to merit publication in Nature Communications.

We thank the reviewer for their consideration of our work. We were disappointed that unlike Reviewer 2, they did not fully recognise the value of our study and the unique insights our experimental results provide into the doodling process. However, we hope the substantial additional experiments within the revision (including further nanopore sequencing and new atomic force microscopy) help them better appreciate our contribution to understanding this intriguing mechanism for DNA synthesis.

One general concern with this type of experimental approach is that the data can be easily skewed by contamination. The authors acknowledge likely contamination of the individual NTP mixes with other NTPs and it seems plausible that, in the absence of a careful analytical characterization of the purity of the NTPs and other ingredients the results are significantly affected by contamination, skewed ratios of NTPs in the mixes, or even contamination with DNA. The authors state that the patterns were "robust and reproducible across replicates", however the replicate data in the supplementary material show readily noticeable differences. It is not clear how to interpret such differences, but in the absence of a well-defined metric, it seems that such a claim of reproducibility cannot be easily made.

The reviewer makes an important point that we agree requires further elaboration. To better understand the scale of contaminants in our reactions, we spoke with New England Biolabs who provided all the key reagents used in our doodling reactions. Regarding contamination levels of the dNTPs, they confirmed that their individual dNTPs and dNTP mixes were ≥99% pure (determined by HPLC). It is therefore unlikely that this contamination could lead to significant amounts of doodled DNA containing these bases or skews in compositions. However, it could potentially account for the recovery of sequences containing unexpected bases due to the sensitivity of the nanopore sequencing itself. This is further supported by the very small amounts of DNA produced by many of these reactions (below the detection limit for the Qubit of 4 ng/μL), which compares to 100s–1,000s ng/μL for many of the other doodling reactions. To highlight these points this section was updated:

"It is unclear how DNA synthesis with bases not included in the reaction mix can take place, but could be due to contaminants in the individual dNTP mixes or polymerase preparations used (the commercial provider of the reagents used here guarantee ≥99% pure dNTPs determined by HPLC) as well as the high sensitivity of the nanopore sequencing."

In terms of the reproducibility of the experiments, we agree that the terminology "robust" may be misinterpreted in relation to the experiments with the limited types of dNTPs. Our intention was to highlight the consistent pattern in dominant A⁴T⁴ and A⁵T⁵ motifs produced. To ensure we are clear,

we have reworded this section to highlight the consistent qualitative features, but increased quantitative variability in the composition of the doodled DNA pools:

“In these cases, we found as expected that sequences were nearly entirely composed of these bases (87% of reads <250 nt and 99% of reads >250 nt). Interestingly, short and long reads were dominated by A⁴T⁴ repeats in addition to A⁵T⁵ repeats at a lower frequency (73% and 26%, respectively, across all reads >65 nt long) (Figure 10c). In addition, while qualitative features of these repetitive sequence motifs were consistent across experimental replicates (Supplementary Figure 13), much greater variation in the absolute fractions of specific motifs was seen compared to prior doodling reactions. Specifically, the A⁴T⁴ and A⁵T⁵ motifs that dominated both reactions saw drops of 16% and 10%, respectively across replicates. This suggests that these reactions may be more sensitive to minor differences in the setup of the reactions and external contaminants.”

In summary, while very low levels of contamination may be present in the dNTPs and enzymes, these are very unlikely to affect the strong and reproducible signals we see. We have updated the results to highlight these facts, as well as explain how the sensitivity of the nanopore sequencing enables very weak signals, potentially produced by contaminants, to be observed when virtually no doodling takes place.

I do believe, a priori, that different polymerases and different environmental conditions can result in reasonably well defined patterns of nucleotide incorporation, but this is neither surprising nor necessarily as useful as the authors suggest. The authors suggest or imply that it might be possible to control the “doodling” activity to the point of reaching something approaching sequence-controlled synthesis. In my opinion this is farfetched, particularly given the existence of polymerases like TdT, which are more efficient for untemplated synthesis and are approaching full controllability. Even the title mentions “control of untemplated DNA polymerase activity” but this is far from what is actually shown.

We thank the reviewer for this useful comment and apologise if we were not clear in the way we presented our findings. Firstly, in response to the comment of well-defined patterns of nucleotide incorporation being neither surprising nor useful, we’d like to stress that until this study only a small number of individually cloned fragments of DNA from doodling reactions had been sequenced (see DOIs: 10.1093/nar/26.20.4652, 10.1007/s00792-014-0706-1, 10.1021/bi0489614). While biases in biomolecular processes are common, precisely what these biases are is typically difficult or impossible to predict *a priori*. Experiments like those presented in our paper are essential to uncover this information and there can still be surprises in the specifics of the biases and the possible implications these might have for the wider biology in which they are embedded. As highlighted by the second reviewer’s comments, understanding these processes and providing the data to back up personal beliefs or intuition is crucial in assessing whether such a phenomenon may have broader impact or use.

In response to the second point on control and the emerging use of TdTs, we would like to stress that we have been incredibly careful when presenting our results to not suggest that we envisage this type of mechanism in the short-term being suitable for sequence perfect synthesis of long DNA fragments. Instead, as mentioned in the Discussion of the original submission, we envisage its value in guiding the synthesis of DNA pools with features in the compositional statistics of the sequences present (see final paragraph of the Discussion). We made a concerted effort to use the term “*guided*” in key points and avoided any reference to terminology like “precise control”. We also disagree that it is inappropriate for the general term “control” to be used, given that the external environmental factors we change do provide us with an ability to alter the outcomes in a reproducible way.

As for the use of doodling for DNA synthesis as being “*farfetched*” and TdTs being a better approach, we do see some specific use-cases that were likely not sufficiently highlighted or elaborated on in the initial submission. Specifically, the ability for doodling DNAPs to produce highly repetitive sequences. Repeats are very difficult to synthesize using current approaches (including TdTs), but

are important structurally in virus genomes (e.g., in terminal repeats) and are present throughout the human genome (e.g., Alu repeats are 11% of the human genome and modify gene expression). Being able to synthesize such sequences would fill a gap in current TdT capabilities and could help us to further understand the effects of these repeats in nature and engineered biological systems. Furthermore, the ability for DNAPs to perform both untemplated and templated synthesis offers yet further interesting possibilities that TdT technologies cannot support (e.g., *de novo* synthesis and amplification in a single reaction). These capabilities in our mind make it foolish to discount DNAP-based enzymatic synthesis as a dead-end that is not worth pursuing.

To try and address the reviewer's concern, we have expanded the Discussion to be more explicit about the value we see in this phenomenon for DNA synthesis and carefully reviewed the entire manuscript to ensure our use of the word "control" is warranted:

"The vertebrate immune system uses TdT to add nucleotides without any template in order to increase antigen receptor diversity³⁵ and was previously considered the only enzyme able to generate DNA without a template. Enzymatic DNA synthesis using TdT has spawned several successful companies such as Molecular Assemblies, DNA Script, and Ansa Biotechnologies³⁶ who have developed specialized protocols, 3'-protection groups³⁷⁻³⁹, and re-engineered TdT with more rapid nucleotide addition (Patent WO/2019/135007), modified catalytic activity⁴⁰, and with dNTPs tethered to the enzyme²⁸. These advances help to mitigate some of the limitations of TdTs, including the bias towards the addition of particular nucleotides⁴¹. Development of DNAP-based enzymatic DNA synthesis could similarly improve the control of untemplated DNA synthesis and provide further benefits. For example, TdT approaches typically struggle to produce highly repetitive and structurally prone sequences that are known to be important structurally in virus genomes (e.g., in terminal repeats) and which are present throughout the human genome (e.g., Alu repeats are 11% of the human genome and modify gene expression). The demonstrated ability for doodling by DNAPs to effectively synthesize repetitive sequences would fill a gap in current TdT capabilities and could help us to further understand the effects of repeat sequences in nature and engineered biological systems.

*Beyond sequence complexity, the yield, purity and length of DNA synthesized by any methodology depends on the efficiency of each cycle of nucleotide addition. Despite the excellent reported efficiency from DNA Script of 99.7%⁴², mathematically, the yield of products of 1 kb in length is only ~5%, and even 99.9% efficiency yields <37% of 1 kb length products³⁶, not taking into consideration the error rate. Longer DNA constructs have been made using DNA assembly methods, namely a 10 kb construct from Ribbon Biolabs using Gibson assembly (Patent WO/2019/073072) and whole genomes using yeast-based DNA assembly^{43,44}. However, assembly of DNA fragments can be laborious and expensive so the development of novel technologies to write longer and more accurate DNA constructs will be vital in advancing synthetic genomics. Furthermore, while the fidelity of doodling at present means it is impossible to synthesise long and defined sequences, the ability of DNAP-based methods to both carry out templated and untemplated DNA synthesis, could open up new avenues for DNA production that exploit *de novo* synthesis and amplification concurrently – features that no existing technologies are able to combine."*

Also missing from the manuscript is an explanation and detailed analysis/interpretation of the histograms obtained from the sequencing.

We apologise to the reviewer for having not provided sufficient analysis/interpretation of the read length histograms obtained from the sequencing. We had initially focused most of our efforts on understanding the patterns in the sequences produced but have attempted to address this concern in the revision by expanding the Results when these are introduced to cover their shapes in more depth. Given that the size distributions capture statistics related to the underlying generative process, we have also made comments relating to the likely rates of synthesis and possible inhibitory effects seen for very long fragments.

Related to this point, we also carried out further experiments to try and understand the discrepancy between the gel images that showed large quantities of very long DNA fragments (10,000s–100,000s nt) to the read length distributions from the nanopore sequencing. To do this, we performed

additional atomic force microscopy (AFM) experiments on the doodled DNA pools to better understand the physical structure of the molecules produced. Interestingly, we discovered highly branched, tree-like structures that could account for the observed discrepancies. Specifically, the large smears seen in the gels that typically would be indicative of wide-ranging lengths of DNA product are likely an artifact of the physical structures within the DNA pools themselves. Moreover, the close correspondence in the median molecule/read lengths from the AFM images and nanopore sequencing (518 nt versus 552 nt, respectively), suggests that the read length distributions are capturing a realistic picture of the true lengths of the doodled DNA molecules. This new experimental data is presented and analysed as a new section of the Results called “*Physical structure of doodled DNA pools*”.

Reviewer #2:

The manuscript from Castle et al describes analysis of untemplated DNA synthesis by DNA polymerases. This is an interesting but poorly understood area, and I think this study is very interesting and presents a new level of insight into the DNA made by this unusual mechanism. I think there a couple points that should be addressed and updated, but the paper is mostly suitable for publication and adds a significant contribution to the understanding of untemplated polymerase activity.

We are grateful to the reviewer for their careful assessment of our work and are pleased that they recognised its valuable contribution, providing the most in-depth picture of DNAP doodling activity to date. Below we have outlined how the minor comments raised have been fully addressed in our revision and some of the additional experiments performed. This includes new nanopore sequencing and atomic force microscopy that provides to our knowledge the first “look” at the structure of the doodled DNA products we synthesise.

1) The presence of G and C bases in the reactions containing only dA/TTP is puzzling, which the authors admit. The suggestion of dNTP contamination seems pretty unlikely, as the G/C-containing sequences are pretty prevalent and a significant amount of the bases would have to be present. The reactions are heated at high temperature for very long times, could some deamination of dA to dI give a signal read as G or C? Nucleotides are easily analyzed by mass spec and the vendor here likely could share any contaminant levels. I feel this is a mysterious result worth a bit more explanation.

The reviewer makes an excellent point about the high temperatures that the reactions are run at and the possibility of deamination of deoxyadenosine (dA) to deoxyinosine (dI) which could influence the nanopore sequencing. To assess whether dI nucleotides would be resolved as G or C bases, we carried out a new experiment with pairs of complementary synthesised primers 43 nt in length that contained a run of three dI nucleotides in their centre on one of the strands. Pools of these oligos were annealed and nanopore sequenced. Analysis of the sequencing data showed that the flanking 20 nt were accurately resolved, while the three dI nucleotides were typically basecalled as C or G bases with the most common resolved sequences being ‘CGC’ that was seen for 32% of all reads for that strand. This contrasted with the opposite strand where most reads (72%) were correctly called as ‘TTT’ although a small fraction (11%) were called as ‘CCC’. This miscalling is likely due to the effect that the complementary strand containing the modified nucleotides has during sequencing.

As recommended by the reviewer, we also spoke with New England Biolabs about contamination levels in the dNTPs used for our experiments. They confirmed <1% contamination (measured by HPLC), which while low, would be measurable with the highly sensitive nanopore sequencing carried out. Even so, it is unlikely this would lead to significant amounts of doodled DNA containing these bases. Furthermore, we carried out a statistical analysis of the *E. coli* BL21(DE3) genome to assess whether, contamination by cells used in the production of the polymerases could act as a source of the repetitive motifs we recovered. We found that only a single motif was found once within the genome meaning it cannot be the source of the diverse repetitive sequences we see.

2) The authors acknowledge that the read length distributions might be biased to shorter products, but I think this is more significant than discussed. From the agarose gels in the paper it would seem the DNA products are overwhelmingly many kb in length, but the sequencing reads are predominantly very short. Could some of the longer DNAs be branched? Single-stranded? Would a HMW-focused library prep give reads more in line with the nature of the products? My concern is that the reads aren't reflective of the real nature of the DNA products and I think this could be examined more.

The reviewer again makes some excellent suggestions about further analysis that could be carried out to better understand the nature of the doodled DNA. To directly address these, we performed two sets of additional experiments making use of (i) atomic force microscopy (AFM) and (ii) nanopore sequencing using a high-molecular weight (HMW) preparation protocol.

Results from the AFM highlighted a fascinating insight into the structure of the doodled DNA, showing a highly branched structure that is likely formed through base pairing of repetitive sequence motifs across different doodled DNA molecules. This leads to tree-like structures that would help explain the large smears seen in the agarose gels because the tree-/mesh-like structures would run significantly slower and give the false impression of very large molecules. We do still see some very long products (~2800 nt) in the AFM data. However, the majority are much shorter (<2,000 nt), which is more in line with the distributions we see from the nanopore sequencing. We added a new section to the Results called “*Physical structure of doodled DNA pools*” to describe this new AFM data and the fascinating structural insights it provides. To our knowledge, this represents the first time that AFM has been performed on doodled DNA and the first comprehensive images and analysis of the branching structures produced, adding further to the novelty and importance of this work.

We also carried out nanopore sequencing experiments that make use of approaches to help ensure the stability of HMW products during the sequencing. These experiments, produced DNA fragment length distributions with a slightly larger median value, further supporting the idea that the very long DNA products seen in the agarose gels are in fact likely DNA nanostructures formed between shorter doodled DNA fragments. This analysis was added as a new Results section called “*Assessing the capture of high molecular weight DNA*”.

3) The Family A and B polymerases used here give different transition frequencies, are these similar to published analysis of the polymerases error spectra when fidelity and misincorporation are being examined?

This is a very interesting question, given that our data spans both Family A (Taq) and B (Vent, Terminator) polymerases. To assess possible biases, we calculated average base transition probabilities to complement the molecule-level transition probabilities shown in the blue panels of Figures 2 and 4 of the original manuscript. In comparison to the known error spectra of these polymerases, we found clear differences. This suggests that the structural constraints for doodling activity are very different to those affecting DNA copying fidelity and that these biases can be shaped through DNAP engineering. A new section call “*Analysis of DNA polymerase transition frequencies*” has been added to present this new data and analysis.

4) This is perhaps a bit pedantic, but I would suggest the authors not state they're using "real-time PCR" to measure the production of the DNA. They're doing isothermal reactions monitored by real-time fluorescence, no PCR is occurring at all just using a "qPCR machine" as a fluorimeter. I think these descriptions should be adjusted accordingly. Although it would be interesting to see what DNA products would be produced if these reactions were thermocycled, the introduction of denaturing steps might give reason to annealing of "primer" strands and different DNA.

We thank the reviewer for this suggestion and agree that “*monitoring of real-time fluorescence*” is a more accurate description of our methodology. We have made appropriate changes throughout the

manuscript and in the figures to clarify this point and be more explicit about how qPCR machines were used.

In terms of adding thermocycling to the reaction, we decided to carry this out for both Taq and Vent, alternating between 65°C for 20 mins and 74°C for 20 min for 9 cycles. We expected this to accelerate the production of the repetitive sequence motifs, but instead it resulted in a complete breakdown in their formation for both polymerases. Overall, virtually no doodling was seen with an order of magnitude less DNA produced, with most of it being mixtures of genomic contaminants from the *E. coli* genome (likely from the commercially provided enzymes we were using). We have added a new section to the Results called "*Impact of cycling thermal conditions*" to discuss these new experiments and analysis.

Response to reviewers (NCOMMS-24-60521A)

Find below our responses (in blue) to the final minor comments from the reviewers. We thank them for their continued and valuable feedback throughout the review process. We have also carefully reviewed the entire manuscript and made minor changes throughout to address additional editorial requests to ensure the format is appropriate for publication.

Reviewer #1:

The revised manuscript has been significantly extended with the addition of new experimental results as well as extensive revisions to the text, in both cases to address concerns from the reviewers. In my opinion, the revisions have significantly improved the manuscript, particularly by addressing the possibility that some of the experimental results may be due to DNA contamination. Given these improvements, I recommend that the manuscript be published in Nature Communications with minor changes. In particular:

We thank the reviewer for their further consideration of our work and their appreciation of our efforts to expand the scope of the work to address their previous concerns.

1. In my view, the value of the “doodling” to DNA synthesis is oversold. For example, the statement (line 496) to the effect that it could be disruptive to current chemical or TdT-based synthesis cannot be justified by the results presented. I suggest that this type of statement, here and elsewhere, be toned down a couple of notches, as there is almost no conceivable path that would allow doodling to reach even the level of control available via current TdT variants or chemical synthesis.

We agree that doodling, as presented, is unlikely to reach the accuracy seen in current TdT technologies, even the medium term. To ensure we do not overstate its possibilities, as suggested by the reviewer, we have toned down statements related to future sequence perfect synthesis of DNA by doodling throughout the manuscript.

2. I think the interest in the author's results will be more in the direction of potential signatures of doodling in natural systems. However, even here, I think the authors need to be more careful with unsupported statements. The statement (lines 530 to 539) regarding hits in nucleotide databases of e.g. CTATAG, needs to be either removed or supported with a statistical analysis. Any 6mer is going to be present in very large numbers in any large genome, and even if the doodling mechanism had some role in early evolution, it seems likely that a few billion years of evolution would have any evidence of this.

We agree that in the short term, the impact of doodling on genome content is an interesting and accessible direction to explore and that careful consideration of confounding factors will be essential. We have expanded the Discussion a little to cover these important points: *“An intriguing future direction for this work would be to perform a detailed analysis of common doodled motifs throughout known genome sequence space, to assess whether doodling might have played a role in fuelling early evolution (e.g., through comparisons of motifs in genes of early origins) acting as a feedstock for natural selection to act upon. Any such analysis would need to employ careful statistical analyses to ensure other evolutionary processes did not confound the results. It may also be more productive to consider using bottom-up synthetic biology approaches in which systems can be more thoroughly observed and precisely controlled over time.”*

In relation to the comment on the analysis we did of the repetitive sequence motifs, we did not search for a 6-mer sequence across the NCBI, but 5 copies of the 6-mer sequences (a 30-mer sequence). This is very unlikely to occur by random chance ($p = 8.67 \times 10^{-19}$). Due to this confusion, we have elaborated on this sentence to highlight our point more clearly: *“A quick BLAST search for the 30-mer sequence (CTATAG)⁵, a long repeat of the motifs we find, across the NCBI core nucleotide database yields large numbers of hits from across the tree of life, with many being perfect matches. Such sequences are*

unlikely to occur by random chance ($p = 8.67 \times 10^{-19}$, assuming all nucleotides are equally probable). However, it remains to be seen if these patterns are the result of prior doodling activity or if they arose through other means.”

Reviewer #2:

The authors have, in my view, done an excellent job responding to reviewer comments and have added a significant amount of new data. These additions result in a more complete and clearer investigation of the interesting activities of the DNA polymerases presented. The AFM data in particular helps clear up one of the biggest questions I had from the first version. My only minor issue is that the branched DNAs were produced by Taq, which given its 5'-3' exonuclease activity does not have strand displacement ability and generally does not make branched DNA like you get in RCA/MDA/etc. So, these branches must not be 5' extensions but rather something else? More a note of curiosity than something the authors need to address.

Overall, I think this is now an impressive collection of data that explores the findings very thoroughly and with more than enough discussion of the various hypotheses and potential applications of the phenomenon. In my opinion this manuscript is suitable for publication as is and serves as an important contribution to this field.

We thank the reviewer again for their careful assessment of the revision and were pleased to see their positive view on its suitability for publication. The comment about the branching structure of the DNA observed by AFM is an interesting one. Our hypothesis is that these are not truly branched molecules, but instead, the branches are in fact caused through base pairing of two separate ssDNA molecules at sequences containing the repetitive motifs we observe in the sequencing. This is supported by some of the raised heights of the molecules at these points in some of the AFM images and is mentioned in the main text: “Closer inspection of some of these imaged molecules showed increased peaks in height at points where branches occurred (Figure 5b, white arrows). This suggests that these regions may be double stranded, supporting the idea of base pairing between separate ssDNA molecules, or through the formation of hairpins within a single ssDNA molecule.”